# Holocene temperature response to external forcing: Assessing the linear response and its spatial and temporal dependence

Lingfeng Wan[1, 2], Zhengyu Liu[3], Jian Liu[1, 2, 4], Weiyi Sun[1, 2], Bin Liu[1, 2]

[1]Key Laboratory for Virtual Geographic Environment, Ministry of Education; State Key Laboratory Cultivation Base of Geographical Environment Evolution of Jiangsu Province; Jiangsu Center for Collaborative Innovation in Geographical Information Resource Development and Application; School of Geography Science, Nanjing Normal University, Nanjing, 210023, China.
[2]Jiangsu Provincial Key Laboratory for Numerical Simulation of Large Scale Complex Systems, School of Mathematical Science, Nanjing Normal University, Nanjing, 210023, China.
[3]Atmospheric Science Program, Department of Geography, Ohio State University, Columbus, OH43210, USA.
[4]Open Studio for the Simulation of Ocean-Climate-Isotope, Qingdao National Laboratory for Marine Science and Technology, Qingdao, 266237, China.

*Correspondence:* Jian Liu (njdllj@126.com); Zhengyu Liu (liu.7022@osu.edu)

**Abstract.** Previous studies show that the evolution of global mean temperature forced by the total forcing is almost the same as the sum of those forced individually by orbital, ice sheet, greenhouse gases and meltwater in the last 21,000 years in three independent climate models, CCSM3, FAMOUS and LOVECLIM. This validity of the linear response is useful because it simplifies the interpretation of the climate evolution. However, it has remained unclear if this linear response is valid on other spatial and temporal scales, and, if valid, in what regions. Here, using a set of TraCE-21ka climate simulations, the spatial and temporal dependence of the linear response of the surface temperature evolution in the Holocene is assessed approximately using correlation coefficient and a linear error index. The results show that the response of global mean temperature is almost linear on orbital, millennial and centennial scales in the Holocene, but not on decadal scale. The linear response differs significantly between the Northern Hemisphere (NH) and Southern Hemisphere (SH). In the NH, the response is almost linear on millennial scale, while in the SH the response is almost linear on orbital scale. Furthermore, at regional scales, the linear responses differ substantially between the orbital, millennial, centennial and decadal timescales. On orbital scale, the linear response is dominant for most regions, even in a small area of a mid-size country like Germany. On millennial scale, the response is still approximately linear in the NH over many regions. Relatively, the linear response is degenerated somewhat over most regions in the SH. On the centennial and decadal timescales, the response is no longer linear in almost all the regions. The regions where the response is linear on the millennial scale are mostly consistent with those on the orbital scale, notably western Eurasian, North Africa, subtropical North Pacific, tropical Atlantic and Indian Ocean, likely caused a large signal-to-noise ratios over these regions. This finding will be helpful for improving our understanding of the regional climate response to various climate forcing factors in the Holocene, especially on orbital and millennial scales.

## 1 Introduction

Long term temperature evolution in the Pleistocene is often believed, and therefore interpreted, to be driven mainly by several external forcing factors, notably, the orbit forcing, the greenhouse gases (GHGs), the continental ice sheets and the meltwater flux forcing. (Here, we treat the coupled ocean-atmosphere system as our climate system, such that Earth orbital parameters, GHGs, meltwater discharge and continental ice sheet are considered as external forcing). Implicit in this interpretation is often an assumption that the response is almost linear to the four forcing factors, that is, the temperature evolution forced by the total forcing combined is approximately the same as the sum of the temperature responses forced individually by the four forcing factors. This linear response, if valid, simplifies the interpretation of the climate evolution dramatically, because each feature of the climate evolution can now be attributed to those on different forcing factors. One example is the global mean temperature evolution of the last 21,000 years (COHMAP members, 1988; Liu et al., 2014). It has been shown that the global mean temperature response is almost linear to the four forcing factors above in three independent climate models (CCSM3, FAMOUS and LOVECLIM, Fig.2 of Liu et al., 2014) with the temperature evolution forced by the total forcing almost the same as the sum of those individually forced by each forcing factor. Furthermore, this deglacial warming response is forced predominantly by the increase of GHGs, with significant contribution from the ice sheet retreat. This linear response, however, has not been assessed quantitatively for the climate evolution in the Holocene. The Holocene period poses a more stringent and interesting test of the linear response, as it removes the deglacial global warming response that is dominated by that to increased $CO_2$ and ice sheet retreating (Figure 2A of Liu et al., 2014). An even more interesting and practical question here is, besides the global mean temperature response on the slow orbital time scale, how linear is the response at shorter temporal scales and smaller spatial scales, throughout the Holocene?

In general, the assessment of the linear response, in principle, can be done in a climate model using a set of experiments that are forced by the combined forcing as well as each individual forcing. Furthermore, each forcing experiment has to consist of a large number of ensemble members. This follows because a single realization of a coupled ocean-atmosphere model could contain strong internal climate variability on a wide range of timescales (Laepple and Huybers, 2014), from daily variability of synoptic weather storms (Hasselmann, 1976), to interannual variability of El Nino (Cobb et al., 2013), interdecadal climate variability (Delworth and Mann, 2000), all the way to millennial climate variability (Bond et al., 1997). The ensemble mean is therefore necessary for suppressing internal variability and then generating the truly forced response to each forcing. The goodness of the linear response can therefore be assessed by comparing the response to the total forcing with the sum of the individual responses. One practical problem with this ensemble approach is, however, the extraordinary computing costs, especially for long experiments in more realistic fully coupled general circulation models. A more practical question is therefore: is it possible to obtain a meaningful assessment of the linear response using only a single realization of each forced experiment for the Holocene, such as those in TraCE-21ka experiments (Liu et al., 2014).

Strictly speaking, it is impossible to disentangle the forced response from internal variability in a single realization. This would make the assessment of linear response difficult. However, it is conceivable that, if our interest is the slow climate evolution

of millennial or longer time scales in response to the slow forcing factors such as the orbital forcing, ice sheet forcing, GHGs and meltwater flux, the assessment is still possible, albeit approximately, at least for very large scale variability. This follows because these forcing factors are of long time scales and of large spatial scales, the forced response signal should therefore also be on long time scales and large spatial scales, if the response is approximately linear. An extreme example is the almost

linear response in the global temperature of the last 21,000 years as discussed by Liu et al., (2014). In contrast, internal variability in the coupled ocean-atmosphere system tends to be of shorter time scales, decadal to centennial, and of smaller spatial scales, at least in current generation of coupled ocean-atmosphere models. This naturally leads to two questions. First, how linear is the climate response at different spatial and temporal scales, quantitatively? Second, in what regions, the linear response tends to dominate? The answer to these questions should help improving our understanding of regional climate

response during the Holocene. A further question is: if the linear approximation is valid, what is the contribution of each forcing factor in different regions and at different time scales. This question will be addressed in a follow-up paper (Wan et al., 2019).

In this paper, we assess the linear response for the Holocene temperature evolution quantitatively, using 5 forced climate simulations in CCSM3 (Liu et al., 2014), with the focus on the spatial and temporal dependence of the linear response. We

will assess the linearity response on the timescales of orbital, millennial, centennial and decadal, and of the spatial scales of global, hemispheric and regional. The data and methodology are given in section 2. The dependence of the linear response on spatial and temporal scales are analysed in section 3. A summary and further discussions are given in section 4.

## 2 Data and Methods

### 2.1 Data

The data is from TraCE-21ka (Liu et al., 2009, 2014), which consists of a set of 5 synchronously coupled atmosphere-ocean general circulation model simulations for the last 21,000 years. The simulations are completed using the CCSM3 (Community Climate System Model version 3). The simulation forced by the total forcing (experiment ALL) is forced by realistic continental ice sheets, the GHGs, orbital forcing and melting water fluxes. The ice sheet is changed approximately once every 500 years, according to the ICE-5G reconstruction (Peltier, 2004). The atmospheric GHGs concentration is derived from the

reconstruction of Joos and Spahni (2008). The orbital forcing follows that of Berger (1978). The coastlines at the LGM were also taken from the ICE-5G reconstruction and were modified at 13.1ka, 12.9ka, 7.6ka, 6.2ka, after which the transient simulation adopted the present-day coastlines. The meltwater flux follows largely the reconstructed sea level and other paleoclimate information and, in the meantime, reconciles the response of Greenland temperature and AMOC strength in comparison with reconstructions. More information on the details of the experiment and forcing can be seen in He (2011) or

the TraCE-21ka website http://www.cgd.ucar.edu/ccr/TraCE/.

The transient simulation under the total climate forcing reproduces many large scale features of the deglacial climate evolution consistent with the observations (Shakun et al., 2012; Marsicek et al., 2018), suggesting a potentially reasonable climate

sensitivity in CCSM3, at global and continental scales. In addition to the all forcing run (ALL), there are four individual forcing runs forced by the orbital forcing (ORB), the continental ice sheets (ICE), the GHGs (GHG) and meltwater forcing (MWF), respectively (Liu et al., 2014, Table 1). In these four experiments, only one forcing varies the same as in experiment ALL, while other forcing/conditions remain the same as at 19ka. Therefore, this set of experiments can be used to study the linear response of the climate to the four forcing factors. Here, we will only examine the surface temperature response in the Holocene (last 11, 000 years).

## 2.2 Assessment Strategy

We will use correlation and normalized RMSE to assess the linear response (see next subsection for details). We note, however, our assessment of linear response is approximate. Before introducing the details of the assessment method, it is useful here to make some general comments on the linear response assessment. As pointed out by one reviewer, strictly speaking, the assessment of linear response requires one to answer two questions.

Q1: How linear is the response to external forcing?

Q2: What is the relative importance of externally forced vs. internal variability, assuming the response were linear?

Specifically, for Q1: If we denote the temperature response to the full external forcing by $T_R(F_{all}(t))$, the response to the individual forcings by $T_R(F_i(t))$ (with $i = 1, \ldots, 4$), and the internal temperature variability of the five model simulations by $T_{I,all}, T_{I,1}, T_{I,2}, T_{I,3}, T_{I,4}$, respectively, the linearity of the response could be defined by the extent to which the total forced response equals the sum of the individual response, or

$$T_R(F_{all}(t)) = \sum_{i=1}^{4} T_R(F_i(t)). \tag{1}$$

In our case, we only have a single member for each experiment as

$$T_{all}(t) = T_R(F_{all}(t)) + T_{I,all}(t),$$

and

$$T_i(t) = T_R(F_i(t)) + T_{I,i}(t),$$

and the linearity is assessed from the correlation (and normalized RMSE) between the sum of the individual experiments $\sum_{i=1}^{4} T_i(t)$ and the total forcing experiment $T_{all}(t)$. Therefore, our linearity assessment is contaminated by the noise of internal variability. This can be seen, for example, in the correlation as

$$\left[cor\left\langle T_{all}(t), \sum_{i=1}^{4} T_i(t)\right\rangle\right]^2 = \frac{[< T_{all}(t),\ \sum_{i=1}^{4} T_i(t) >]^2}{< T_{all}(t),\ T_{all}(t) >< \sum_{i=1}^{4} T_i(t),\ \sum_{i=1}^{4} T_i(t) >}$$

$$= \frac{[< T_R(F_{all}(t)),\ \sum_{i=1}^{4} T_R(F_i(t)) >]^2}{\{Var[T_R(F_{all}(t))] + Var[T_{I,all}(t)]\}\{\sum_{i=1}^{4} Var[T_R(F_i(t))] + \sum_{i=1}^{4} Var[T_{I,i}(t)]\}} \tag{2}$$

Here, $< >$ indicates covariance, and we have assumed that the forced response $T_R(F_*(t))$ and internal variability $T_{I,*}$ (where *=all, 1, 2, 3, 4) are independent of each other; in addition, the time series is sufficiently long so sampling errors are negligible.

The correlation therefore depends on the single/noise ratio. If the noise (internal variability) is large, the correlation will be much smaller than 1 even if the response is purely linear as in (1). The only way to suppress internal variability is to perform a large number of ensemble simulations for each experiment. Given only one member for each experiment, we have to be content that our linear assessment using (2) is approximate, depending on the signal/noise ratio. Related to this problem of

single member experiment, since we can't distinguish internal variability from forced response clearly, Q2 can't be assessed exactly either.

In spite of these potential issues, with a single member for each experiment, useful information can still be extracted on linear response. Our general hypothesis is that the slow (orbital and millennial) and large (continental and basin) variability is composed mostly of forced signal and the faster (centennial and shorter) and smaller variability is mostly associated with

internal variability of noise. In other words, in our set of single member of simulations, the signal over noise ratio is large for slow variability but small for faster variability. Qualitatively, this hypothesis seems reasonable. First, all the four external forcing factors are of slow time scales and large spatial scales; additionally, internal variability is usually weak in the coupled ocean-atmosphere system at slow time scales and large spatial scales. Our focus here is indeed the slow variability and large scale here, so we can roughly treat the slow and large variability in the single realization roughly as the signal and the linearity

of the response may be assessed using eqn. (2). Second, again, because our forcing factors are of slow time scales and large spatial scales, higher frequency or small scale variability in the model should not be dominated by the forced variability (unless the response is highly nonlinear!). Therefore, high frequency or small scale variability can be treated roughly as "noise". This is consistent with later assessment that slow variability seems to be approximately linear response while high frequency variability not. Based on this hypothesis, the signal over noise ratio is also estimated using the variance of slow variability as

signal and of high frequency variability as noise (as in late Fig.7). It should be noted however that this hypothesis is qualitative in nature. One major purpose of this paper is to give a somewhat more quantitative assessment on this hypothesis. How slow, how large and how good will be the linear response?

Our experimental design is proper for linear response assessment here. Alternatively, in another experimental setting, individual forcing experiments are often superimposed sequentially one-by-one, for example, first the ice sheet, second the

ice sheet plus orbital forcing, third the ice sheet, orbital and GHGs, and finally, applying all four forcing of ice sheet, orbital, GHGs and melting water. In this experimental design, the full forcing response is by default the response of the sum response after adding the four forcing factors together, and therefore can't be used to assess linearity of the response. Nevertheless, it should be kept in mind that our four individual forcing experiments are not designed optimally for the study of the linear response in the Holocene. This is because, except for the variable forcing, all the other three forcing factors are fixed at the

19ka condition. As such, the mean state is perturbed from the glacial state, instead of a Holocene state. This may have contributed to some unknown deterioration on the linear response discussed later. Nevertheless, we believe, our major conclusion should hold reasonably well. This is because, partly, the response is indeed almost linear for orbital and millennial variability as will be shown later.

## 2.3 Methods

We use two indices to evaluate the linear response: the temporal correlation coefficient $r$ and a normalized linear error index $L_e$. The correlation coefficient is calculated as

$$r = \frac{\frac{\sum_{t=1}^{n}(S_t-\overline{S_t})\times(T_t-\overline{T_t})}{n}}{\sqrt{\frac{\sum_{t=1}^{n}(S_t-\overline{S_t})^2}{n}}\sqrt{\frac{\sum_{t=1}^{n}(T_t-\overline{T_t})^2}{n}}} = \frac{cov(S_t,T_t)}{\sigma(S_t)\sigma(T_t)} \tag{3}$$

Here, $S_t = \sum_{i=1}^{4} T_i/4$ is the linear sum of the temperature time series $T_i$ of the four single forcing experiments, $T_t$ is the full temperature time series in the ALL run, both at time $t$, and $n$ is the length of the time series. The overbar represents the time mean. The correlation coefficient represents the similarity of the temporal evolution between the sum response and the ALL response. However, the correlation does not address the magnitude of the response. Indeed, even if $S_t$ and $T_t$ has a perfect correlation $r$=1, the two time series can still differ by an arbitrary constant in their magnitudes. Therefore, we will also use a normalized linear error index $L_e$ to evaluate the magnitude of the linear response. Here, $L_e$ is defined as the root mean square error (RMSE) of the sum temperature response from the full temperature response divided by the standard deviation of the full temperature response in the ALL run:

$$L_e = \frac{RMSE}{STD} = \frac{\sqrt{\frac{\sum_{t=1}^{n}[(S_t-\overline{S_t})-(T_t-\overline{T_t})]^2}{n}}}{\sqrt{\frac{\sum_{t=1}^{n}[(T_t-\overline{T_t})]^2}{n}}} = \frac{\sigma(S_t-T_t)}{\sigma(T_t)} \tag{4}$$

In general, a large $r$ (close to 1) and a smaller $L_e$ (close to zero) represents a better linear approximation, with $r$=1 and $L_e = 0$ as a perfect linear response. Therefore, if $r$ is close to 1 and $L_e$ is close to zero, we can conclude that the response is close to linear. As noted above, with a single realization here, our assessment of linear response has limitations. First, if $r$ is sufficiently small and $L_e$ is sufficiently large, we can't confirm the response is either linear or nonlinear, because the small $r$ or large $L_e$ can also be contributed by strong internal variability. Second, if the forcing is dominated by shorter time scale variability, say interannual to interdecadal variability, as in the case of volcanic forcing or solar variability, it will be difficult to assess the linear response. This because the time scales of forced response now overlap heavily with strong internal climate variability in the coupled system, and it will be difficult to separate the forced response from internal variability without ensemble mean. But how to assess the goodness of the linear response from the value of $r$ and $L_e$? We can test the goodness of the linear response statistically on $r$ and $L_e$.

The statistical significance of $r$ for a particular timescale is tested using the Monte Carlo method (Kroese, 2011 and 2014; Kastner, 2010; Binder, 1997) with 1,000,000 realizations on the corresponding red noise in the AR(1) model (autoregressive model of order 1) which uses the AR(1) coefficient derived from the model to generate time series. The fit is use the lag-1 auto-correlation coefficient.

The statistical significance of the $L_e$ of a particular timescale is tested using a bootstrap method (Efron, 1979, 1993) with 1,000,000 realizations on the corresponding time series. Specifically, the bootstrap is done as follows, taking the global mean

temperature as example. First, we will derive the $L_e$ from one random realization on the temperature of the ALL run of the 100-binned data (110 points of data, each representing a 100-yr bin). For this random realization, the order of the original temperature time series is swapped randomly. Then, this realization is used as a new ALL response for comparison with the sum response of the four individual experiments to derive a $L_e$ in eqn. (2). Since the random realization distorts the serial

correlation time with the sum response, one should expect usually a large error $L_e$. Second, we repeat this process for 1,000,000 times on 1,000,000 random realizations; this will produce 1,000,000 random values of $L_e$, forming the PDF of the $L_e$. Third, the minimum 5% level is then used as the 95% confidence level.

Statistical significance of the $L_e$ of a particular timescale is tested using a bootstrap method (Efron, 1979, 1993) with 1,000,000 realizations on the corresponding time series. Specifically, the bootstrap is done as follows, taking the global mean temperature

as example. First, we will derive the $L_e$ from one random realization on the temperature of the ALL run of the 100-binned data (110 points of data, each representing a 100-yr bin). For this random realization, the order of the original temperature time series is swapped randomly. Then, this realization is used as a new ALL response for comparison with the sum response of the four individual experiments to derive a $L_e$ in eqn. (2). Since the random realization distorts the serial correlation time with the sum response, one should expect usually a large error $L_e$. Second, we repeat this process for 1,000,000 times on 1,000,000

random realizations; this will produce 1,000,000 random values of $L_e$, forming the PDF of the $L_e$. Third, the minimum 5% level is then used as the 95% confidence level.

The dependence of the linear response on spatial and temporal scales will be studied by filtering the time series in different scales. For the spatial scale, we will divide the globe into 9 succeeding cases, denoted by 9 division factors: *f=0, 1, 2, 3, 4, 6, 8, 12* and *24* from the largest global scale to the smallest model grid scale. The *f=0* case is for global average while the *f=1*

case is for hemispheric average in the NH and SH. Further division will be done within each hemisphere. Note that each hemisphere has $96(lon.) \times 24(lat.)$ grid boxes, with a ratio of 4:1 between longitude span and latitude span. We divide each hemisphere into $f \times f$ sections of equal latitude and longitude spans, with each area containing the same number of $(96/f) \times (24/f)$ grid boxes, maintaining the ratio of 4:1 between longitude span and latitude span. For example, *f=2* is for the $2 \times 2$ division, with each area containing $48 \times 12$ grid boxes; *f=24* is for the $24 \times 24$ division with each area containing $4 \times 1$ grid

boxes, about the size of $15°(lon.) \times 3.75°(lat.)$, like the size of a mid-size country of Spain or Germany in the mid-latitude. To the time scale, we decompose a full 11,000-yr annual temperature time series (from 11 ka to 0 ka) in 100-yr bins (a total of 110 data bins, or points, each representing a 100-yr mean) into three components. The three components are to represent the variability of, roughly, orbital, millennial and centennial timescales. Following Marsicek et al. (2018), we derive the orbital and millennial variability using a low-pass filter called the locally weighted regression fits (Loess fits) (Cleveland, 1979). First,

the orbital variability is derived by applying a 6500-yr Loess fit low-pass filter onto the temperature time series, and therefore contains the trend and the slow evolution longer than ~6500 years. Second, we apply a 2500-yr Loess fit low-pass filter onto the temperature time series; then, we derive the millennial variability using this 2500-yr low-pass data subtracting the 6500-yr low-pass data. Finally, centennial variability is derived as the difference between the 100-yr binned temperature time series

and the 2500-yr low-pass time series. In addition, we also derive a decadal variability time series. First, we compile the 10-yr bin time series from the original 11,000-yr annual time series (of a total of 1,100 data points, each representing a 10-yr mean). Second, we apply a 100-yr running mean low-pass filter on the time series of the 10-yr binned data. Finally, decadal variability is derived by using the 10-yr binned time series minus its 100-yr running mean time series.

Given the different degrees of freedom especially among the filtered variability of different time scales, it is important to test the goodness of the linear response statistically on $r$ and $L_e$ on different time scales differently. As a reference, the significance level is tested against the global mean temperature series in the ALL run. For the total, orbital, millennial, centennial and decadal temperature time series, the 95% confidence levels are found to be 0.72 (with the AR(1) coefficient 0.96), 0.76 (0.97), 0.65 (0.95), 0.21 (0.31) and 0.19 (0.06), respectively.

In this paper, this AR(1) test for global mean temperature is also used as the common significant test for different spatial scales and in different regions as well. This use of a common significance level is for simplicity here. First, the use of different regional AR(1) coefficient for different regions will make the comparison of the linear responses among different spatial scales (e.g. Fig.3 and 4) and different regions (Fig.5, 6) difficult. Second, except for the orbital time scale, the AR(1) coefficient for the global mean temperature is larger than most of the regional AR(1) coefficients (not shown), likely caused by the further

suppression of internal variability in the global mean. As a result, the global mean AR(1) test serves actually as a more stringent test than the local AR(1) test. At the orbital scale, the global mean AR(1) coefficient is in about the middle of the regional AR(1) coefficients. The uncertainly of using the global mean AR(1) coefficient is therefore about the average of those of regional AR(1) coefficients. Third, and, most importantly, as our first study here, our focus is on the global features of the linear response. The difference among the AR(1) coefficients among different spatial scales and different regions are much

smaller than that between different time scales here. Therefore, the global mean AR(1) can still provide an approximate guideline of the significant test properly at different time scales. In later studies, if one's focus is for a specific spatial scale and on a specific region, the regional AR(1) should be used for re-examination of the significance test.

    As a reference, the significance level of $L_e$ is tested against the global mean temperature series in the ALL run. For the total, orbital, millennial, centennial and decadal temperature time series, the 95% confidence levels are found to be 1.23, 1.23, 1.21,

1.24 and 1.36, respectively. This suggests that when the RMSE is less than about 1.2-1.3 times of the total response, the linear sum is not significantly different from the total response at the 95% confidence level. As for the $L_e$ test, since our focus here is on the global feature of the linear response, for simplicity, the significance level derived from the global mean temperature is used as the common confidence level for all regional scales.

## 3 Results

### 3.1 Linear responses at different temporal scales

The global mean temperature provides a useful example to start the discussion of the dependence of the linear response on timescales. We first examine the linear response of the global mean temperature based on its components of orbital, millennial, centennial and decadal variability (Fig.1). Fig.1a is the total variability of global surface temperature derived from the ALL run and the sum of the four individual forcing experiments. The global temperature response is almost linear on the orbital and millennial scales throughout the Holocene (Fig.1b and 1c). The orbital scale evolution is characterized by a warming trend of about 1℃ from 11ka to ~4.5ka before decreasing slightly afterwards. This feature is captured in the linear sum albeit with a slightly smaller magnitude and an additional local minimum around 3ka (Fig.1b). The total variability is very similar to the orbital variability ($r$=0.99, Fig.1a vs Fig.1b). The millennial variability shows 5 major peaks around ~9.8ka, 7.8ka, 4.7ka, 3.7ka and 1.8ka. All these peaks seem to be captured in the sum response albeit with a slightly larger amplitude (Fig.1c). For orbital and millennial variability, the correlation coefficients between the sum and the full responses are $r$=0.83 and 0.71, respectively, both significant at the 95% confidence level and explaining over 50% of the variance; the linear errors are $L_e$=0.63 and 0.92, respectively, also significant at the 95% confidence level. It should be noted, however, that the goodness of the linear response is based on the entire period and is meant for the response of the time scale to be studied. Therefore, even for a good linear response at long time scales, the sum response may still differ from the total response significant at some particular time. For example, for the orbital scale response in Fig.1b, even though the linear response is good in terms of $r$ and $L_e$, there is a 1℃ difference between the sum and total responses at 11ka and 3ka. Therefore, for the orbital scale response, the linear response mainly refers to the trend-like slow response of comparable time scale of the orbital scale, instead of some response features of shorter time scales. Further down the scale, at centennial time scale, the global centennial variability appears also to exhibit a modest linear response (Fig.1d), with $r$=0.44 and $L_e$=1.21. But the linear response of the decadal variability becomes poor (Fig.1e), with $r$=-0.02 and $L_e$=1.99, which is not statistically significant at the 95% confidence level. The result of the analysis on global mean temperature is qualitatively consistent with previous hypothesis that the linear response tends to degenerate at shorter temporal scale, because of the smaller forced response signal and the presence of strong internal variability. It shows that, for global temperature, the response is approximately linear at orbital and millennial timescales, but becomes much less so at centennial scales and fails completely at decadal scales.

### 3.2 Linear responses at different spatial scales

In order to assess the linear response at different spatial scales, we first analyse the linear response on the hemisphere scale for the NH and SH ($f=1$). It is interesting that the linear response significantly differs between the NH and SH (Fig.2). Fig.2a and 2f are the total variability of hemispheric surface temperature of the ALL response and sum response for the NH and SH, respectively. Their components on the 4 timescales (i.e. orbital, millennial, centennial and decadal) are shown in Figs.2b-2e and Figs.2g-2j, respectively. In the NH, the response is almost linear at the millennial scale ($r$=0.82, $L_e$=1.01, Fig. 2c), but not

so strong on orbital ($r$=0.55, $L_e$=0.84, Fig.2b) and centennial ($r$=0.32, $L_e$=2.29, Fig.2d) scales, while only $L_e$ is significant on the orbital scale, and only $r$ is significant on the centennial scale. At the decadal scale, the linear response fails completely ($r$=-0.04, $L_e$=1.99, Fig.2e). In comparison, in the SH, the linear response is dominant at the orbital scale ($r$=0.92, $L_e$=0.43, Fig.2g), but poor on all other timescales, including millennial ($r$=-0.12, $L_e$=2.32, Fig.2h), centennial ($r$=0.14, $L_e$=3.19, Fig.2i) and decadal ($r$=0.03, $L_e$=2.07, Fig. 2j) scales. The linear response of the global mean temperature discussed in Fig.1 therefore seems to be dominated by the SH response on the orbital scale, but by the NH response on the millennial scale. This suggests that the goodness of the linear response depends on both the region and time scale, highlighting the need to study the linear response at regional scales.

The linear response at different spatial scales and on the orbital, millennial, centennial and decadal timescales are summarized in Fig.3 and Fig.4 in the correlation coefficient and linear error index, respectively. Fig.3a shows the correlation coefficients of the orbital variability in each region for the 9 division factors. The cases of global mean ($f$=0) and hemispheric mean ($f$=1) have been discussed in details before. The correlation coefficients for succeeding division factors ($f$=2, 3, ..., 24) show several features. First, as expected, the correlation coefficient tends to decrease towards smaller area (larger $f$). Quantitatively, however, the correlation coefficient does not decrease much, such that even at the smallest area ($f$=24), the correlation in most regions are still above 0.8, statistically significant at 95% confidence level. This suggests that the response at the orbital scale is almost linear over most regions, even at the smallest scale of about a mid-size country like Germany ($f$=24, 15°($lon.$) × 3.75°($lat.$)). Second, the linear response in the NH is slightly better than SH (for $f$≥3), a topic to be returned later. Third, subareas in both hemispheres show comparable linear response across all the spatial scales, with the median correlation all above ~0.8, except that of the NH mean temperature ($f$=1). The linear response of NH is not better than those of regional variability at smaller spatial scales ($f$≥2). This is opposite to the expectation that the linear response becomes more distinct for a larger area, because the average over a larger area tend to suppress internal variability more. This case, however, seems to be a special feature and should be treated with caution. The correlation coefficient therefore shows that, for orbital scale evolution, temperature response is dominated by the linear response over most of the globe, even at regional scales. These features are also consistent with the linear error analysis in Fig.4a.

Millennial variability also shows a weaker linear response for smaller scales, in both the correlation (Fig.3b) and linear error (Fig.4b). Quantitatively, for millennial variability, the response is still approximately linear in the NH over many regions, albeit less so than at the orbital scale. The correlation coefficients remain above 0.6 across most regions even at the smallest division area ($f$=24), contributing to ~40% of the variance. In contrast to orbital variability, where regions in both hemispheres show comparable linear response, millennial variability shows that the response can't be confirmed linear in most regions in the SH, with the median no longer significant at 95% confidence level. Similar to the orbital variability, nevertheless, the responses are more linear in the NH than SH on all the spatial scales.

In contrast to orbital and millennial variability, almost no response can be confirmed linear for centennial variability. The median linear response is no longer significant on the centennial timescale in either hemisphere across spatial scales ($f$>3,

Fig.3c and Fig.4c), with few correlation coefficients larger than 0.3 and contributing less than 10% of the variance. Finally, decadal variability exhibits absolutely no linear response over any spatial scales in either the NH or SH (Fig.3d and Fig.4d). The approximate linear response at the orbital and millennial scales suggest that these two groups of variability are generated predominantly by the external forcing. In contrast, the poor linear response of centennial and decadal variability suggest that

these two groups of variability are caused mainly by the internal coupled ocean-atmosphere processes. This is largely consistent with our original hypothesis. It should be kept in mind that, in our single realization here, the poor linear response on centennial and decadal variability may also be contributed by nonlinear responses of the climate system. But, given the almost absence of forcing variability at this short time scale in our experiments, we do not think that the nonlinear response is the major cause of the poor linear response here.

**3.3 Pattern of the Linear Responses**

We now further study the pattern of the linear response. Fig.5 shows the spatial patterns of the correlation coefficients at orbital (Fig.5a1-a3) and millennial (Fig.5b1-b3) scales for three representative spatial scales, $f=3, 6$ and $24$ (the other factors are not shown because they are similar to the above mentioned three representative spatial scales). For orbital variability (Fig.5a1-a3), the response is almost linear in most regions in the NH on all three spatial scales, with the correlation coefficients above 0.8.

In the SH, the response is also almost linear over the continents, except for over Australia, but is not linear over the ocean. This leads to the significantly reduced linear response in the SH as discussed in Fig.3a-4a. This suggests that orbital variability is likely forced predominantly by external forcing over continents. The overall poorer linear response over ocean than land, however, is puzzling. The orbital time scale is so long that one would expect a similar quasi-equilibrium response over both land and ocean surface. This issue deserves further study in the future. More specifically, at the regional scales, e.g. $f=6$ and

$24$ (Fig.5a2 and 5a3), the response is almost linear over the western half of the Eurasian continent, Northern Africa, Central and South America, most NH oceans and SH tropical oceans, and the Antarctica Continent, as seen in the Fig.5a2 and 5a3 (only those significant correlation coefficients are shown). But the linear response is poor over the North America continent and the eastern Eurasian continent, and the entire Southern Ocean. Similar features can also be seen in the map of linear error (not shown).

For millennial variability, in the NH, linear response shows a similar feature to that of orbital variability, but the linear response is poor over almost all the SH. Fig.5b1-b3 show that the response is almost linear in most regions in the NH at the three spatial scales, with the correlation coefficient above 0.6. At the regional scale, e.g. $f=6$ and $24$ (Fig.5b2 and 5b3), the response is almost linear over the northwestern Eurasian continent, Northern Africa, northern North America, northern Pacific Ocean, southern North Atlantic Ocean and western Arctic Ocean, as seen in the Fig.5b2 and 5b3 (only those significant correlation

coefficients are shown). But the linear response is poor over the southern North America, the eastern and southern Eurasian continent. While in the SH, the linear relationship is poor over almost the entire SH as seen in the correlation map (Fig.5b1-b3). Interestingly, over the NH, the regions of linear response for millennial variability are mostly consistent with the regions of linear response for orbital variability, notably western Eurasian, North Africa, subtropical North Pacific, tropical Atlantic

and Indian Ocean. These preferred region of linear response suggests potentially some common mechanisms of the climate response in these regions, in this model.

In order to understand the cause for the preferred regions of linear response, we examine the signal-to-noise ratio. As discussed in subsection 2.2, our forcing factors are on millennial and orbital time scales, and the linear response is also largely valid for orbital and millennial variability. We will therefore use the variance of the orbital and millennial variability as a crude estimate of the linear response signal. Similarly, since there is no centennial and decadal forcing in our model and the response of centennial and decadal variability are not linear response, we use the variance of the sum of the centennial and decadal variability as a rough estimate for internal variability as the linear noise. Admittedly, this estimation is crude, limited by the single realization here. This signal-to-noise ratio is not directly to address Q2 in subsection 2.2, because the time scales of the signal and noise are different. Instead, it is used as a rough estimation of the relative magnitude of signal-to-noise ratio between different regions, with the assumption that the relative noise level between different regions may be not too sensitive to the time scales. Indeed, the use of signal to noise ratio here is to shed some light on the regional preference of linear response. Figure 6 shows the signal-to-noise ratio for orbital and millennial variability for the three representative spatial scales ($f=3, 6$ and $24$). For orbital variability (Fig.6a1-a3), the signal-to-noise ratio is large (above $10^{(0.6)}$, the log base 10 is taken on the signal to noise ratios) in most regions in the NH. In the SH, the signal-to-noise ratio is also large over the continents, but is small over the ocean. At different regional scales, e.g. $f=6$ and $24$ (Fig.6a2 and 6a3), the signal-to-noise ratio is large over the western half of the Eurasian continent, Northern Africa, Central and South America, most NH oceans and SH tropical oceans, and the Antarctic Continent. But the signal-to-noise ratio is small over Canada and the eastern Eurasian continent, and the entire Southern Ocean. These spatial features of signal-to-noise ratio on the orbital scale (Fig.6a1-a3) are similar to those in the correlation map (Fig.5a1-a3). The spatial correlation between the map of the signal-over-noise ratio in Fig.6 and the corresponding correlation coefficient in Fig.5 is 0.53, 0.58 and 0.49 for $f=3, 6$ and $24$ respectively, as seen in the scatter diagram in Fig.7, all significant at 95% confidence level.

For millennial variability, the signal-to-noise ratio also shows a similar feature to that of orbital variability although overall somewhat smaller (note the different color scales). Fig.6b1-b3 show that the signal-to-noise ratio is large in most regions in the NH in all three spatial scales, with the signal-to-noise ratio above $10^{(-1)}$ (the log base 10 is taken on the signal to noise ratios). At the regional scale, e.g. $f=6$ and $24$ (Fig.6b2 and 6b3), the signal-to-noise ratio is large over the northwestern Eurasian continent, Northern Africa, central North America, northern North Pacific Ocean, Northern Atlantic Ocean and the Arctic Ocean. But the signal-to-noise ratio is small over the North America continent outside the central North America, the South America and the eastern and southern Eurasian continent. While the signal-to-noise ratio is small over almost the entire SH. Over the NH, interestingly, the regions of large signal-to-noise ratio for millennial variability, are mostly consistent with the regions of large signal-to-noise ratio for orbital variability, notably northwestern Eurasian, North Africa, subtropical North Pacific, Northern Atlantic and tropical Northern Indian Ocean. These features of spatial pattern of signal-to-noise ratio on the millennial scale (Fig.6b1-b3) are similar to those in the correlation map (Fig.5b1-b3). The correlation coefficient between the

maps of signal-to-noise ratio and correlation is 0.65, 0.51 and 0.45 for *f=3, 6* and *24*, respectively (Fig.7), again all significant at 95% confidence level.

## 4 Summary and Discussions

In this paper, the linear response is assessed for the surface temperature response to orbital forcing, GHGs, meltwater discharge and continental ice sheet throughout the Holocene in a coupled GCM (CCSM3). The global mean temperature response is almost linear on the orbital, millennial, and even centennial scales throughout the Holocene, but not for decadal variability (Fig.1). Furthermore, the sum response account for over 50% of the total response variance for orbital and millennial variability. Further analysis on regional scale suggests that the response is approximately linear on the orbital and millennial scales for most continental regions over the NH and SH, with the sum response explaining over about 50% of the total response variance. However, the linear response is not significant over much of the ocean, especially over the ocean in the SH. There are specific regions where linear response tends to be dominant, notably the western Eurasian continent, North Africa, the central and South America, the Antarctica continent, and the North Pacific. The strong linear response is interpreted as the region of large signal-to-noise ratio. That is, in these regions, either the orbital and millennial response signal is large, or the influence of the centennial and decadal variability noise is small, or both. This suggests that the orbital and millennial variability in these regions are relatively easy to understand. This finding lays a foundation to our further understanding of the impacts of different climate forcing factors on the temperature evolution in the Holocene of orbital and millennial time scales. This understanding is our original motivation of this work. Further work is underway in understanding the contribution of different forcing factors on the temperature evolution (Wan et al., 2019).

It should be kept in mind that since there is only one member for each experiment, we can't separate the forced response signal from the internal variability of noise clearly at each time scales. Therefore, we can't address Q1 and Q2 raised in subsection 2.2 accurately. Instead, our assessment is likely contaminated by internal variability (see discussions in section 1 and 2). In particular, for smaller scale variability, of which internal variability is likely to be strong and the forced signal is likely to be weak, our correlation may underestimate the linearity of the response (see eqn. (2)). Nevertheless, we speculate that our results on large scale variability still remains robust. Furthermore, at regional scales, although the absolute value of linear correlation of the forced response may be underestimated, it is possible that relative between different regions, the linear assessment may still be somewhat valid. These speculations, however, require much further studies, especially with ensemble experiments. In spite of its limitation, our study represents the first systematic assessment of linear response for the Holocene and can serve as a starting point for further studies in the future.

There are many further issues that need to be studies. Our study here is carried out for a single variable (surface temperature) in a single model (CCSM3) for the Holocene. Yet, the linear response could differ for different variables, in different models, for different periods and for different sets of forcing factors. For example, if we evaluate the precipitation response in the Holocene in CCSM3, the response is less linear than temperature (not shown); this is expected because precipitation response

contains more internal variability and exhibits more nonlinear behaviour than temperature. The assessment will be also different if a different period is assessed, e.g. the last 21,000 years; with a large amplitude of climate forcing, the linear response may degenerate in the 21,000-year period. In addition, the assessment of linear response using only one realization will be difficult to perform for volcanic forcing and solar variability forcing; these forcing factors have short time scales and therefore

their impacts will be difficult to separate from internal variability without ensemble experiments. Finally, it is also important to repeat the same assessment here in different models and to establish the robustness of the assessment. It should also be kept in mind that our assessment is implicitly related to the assumption that, at millennial and orbital time scales, internal variability is not strong relative to the forced responses. Although this seems to be consistent in our model, there is a possibility that internal variability is severely underestimated in the model than in the real world (Laepple and Huybers, 2014). If true, the

relevance of our model assessment to the real world will be limited. It should also be kept in mind that, if the response is dominated by that to a single forcing, the assessment of linear response here becomes one that is more relevant to the question of the forced response vs internal variability, as discussed in Q2 in subsection 2.2. As a further step, though, one can examine if the magnitude of the total response responds to the magnitude of this single forcing linearly.

Even in the context of this model assessment, much further work remains. Most importantly, the purpose of testing the linear

response is for a better understanding of the physical mechanism of the climate response. It is highly desirable to understand why response tends to be linear in some region, but not in other regions. In particular, it is unclear why the linear response is preferred over land than over ocean for orbital and millennial variability. At such a long time scale, one would expect that the upper ocean response has reached quasi-equilibrium and therefore the surface temperature response over land and over ocean should not be too much different. Ultimately, we would like to assess and understand the physical mechanism of the climate

evolution in different regions. These work are underway (Wan et al., 2019).

**Acknowledgments**

This work was jointly supported by the National Key Research and Development Program of China (Grant No. 2016YFA0600401), the National Natural Science Foundation of China (Grant No. 41420104002, 41630527), the Program of Innovative Research Team of Jiangsu Higher Education Institutions of China, and the Priority Academic Program

Development of Jiangsu Higher Education Institutions (Grant No. 164320H116), and US NSF P2C2. We used the output from the full TraCE-21ka simulation, available at https://www.earthsystemgrid.org/project/trace.html.

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

Table 1. TraCE-21ka simulation experiments

| No. | experiment | Forcing | time | Resolution(lat×lon) |
|---|---|---|---|---|
| 1 | ORB | Orbital forcing | 11000 | 48×96 |
| 2 | GHG | GHGs forcing | 11000 | 48×96 |
| 3 | ICE | Ice sheets forcing | 11000 | 48×96 |
| 4 | MWF | Meltwater forcing | 11000 | 48×96 |
| 5 | ALL | Orbital + GHGs + ICE sheets + Meltwater forcing | 11000 | 48×96 |

# global annual mean temperature (°C)

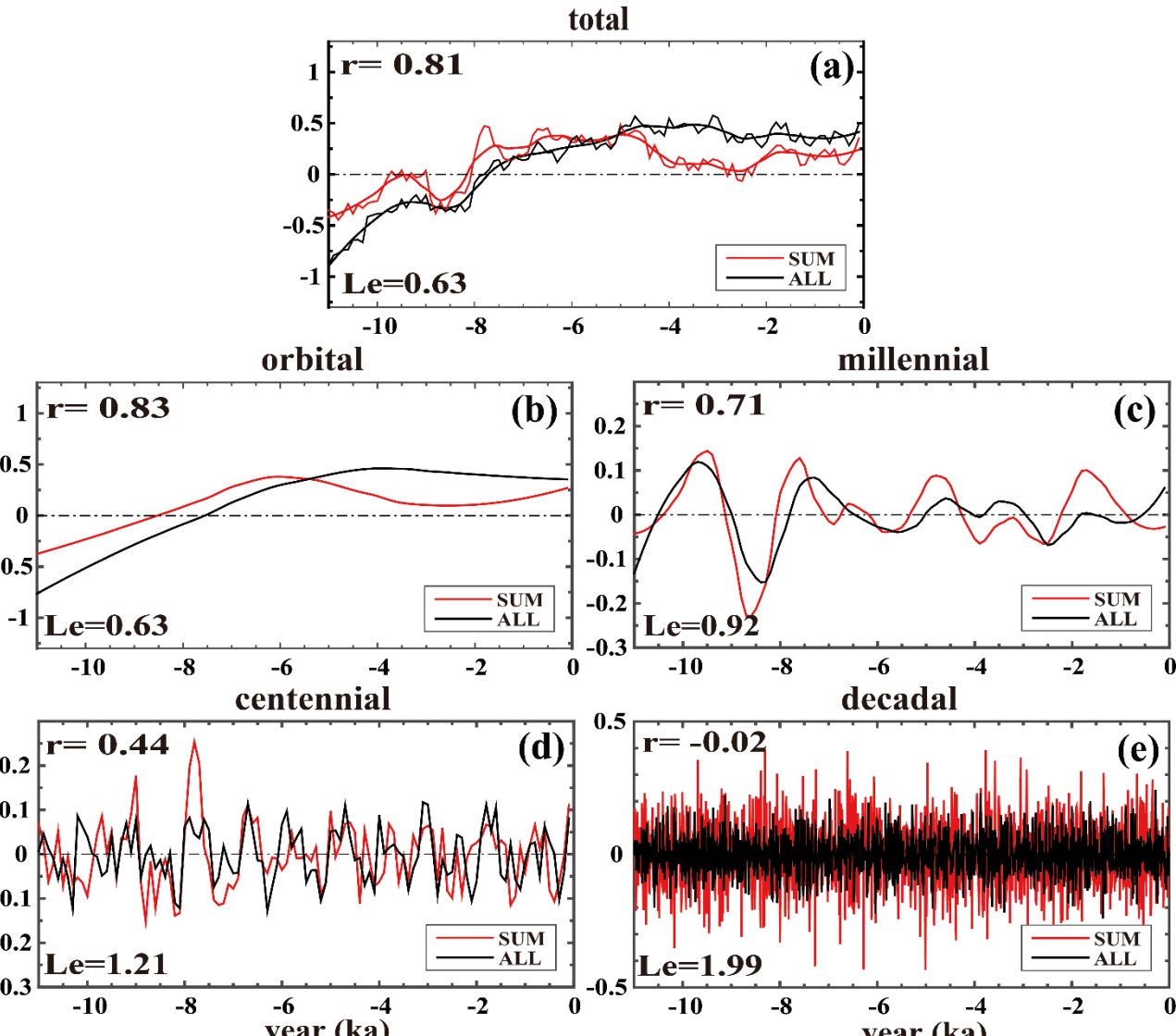

Figure 1: The global annual mean surface temperature time series derived from the ALL run (black) and the SUM (the sum of four single forcing run, red). In the (a), the thin line is the 100-yr binned time series, the thick line is 2500-yr loess fitted time series. The orbital scale variability (b) is represented by the 6500-yr loess fitted series. The millennial variability (c) is represented by the 2500-yr loess fitted data subtracting the 6500-yr loess fitted data. The centennial variability (d) is represented by the 100-yr binned data subtracting the 2500-yr loess fitted data. The decadal variability (e) is represented by the 10-yr binned origin time series subtracting its 100-yr running mean. The correlation coefficient (r) is given at the upper left corner and the linear error ($L_e$) is given at the lower left corner of each panel. The x axis is year (ka, 0 is AD 1950, negative is before AD 1950), and the y axis is temperature anomaly (°C, relative to 11ka-0ka).

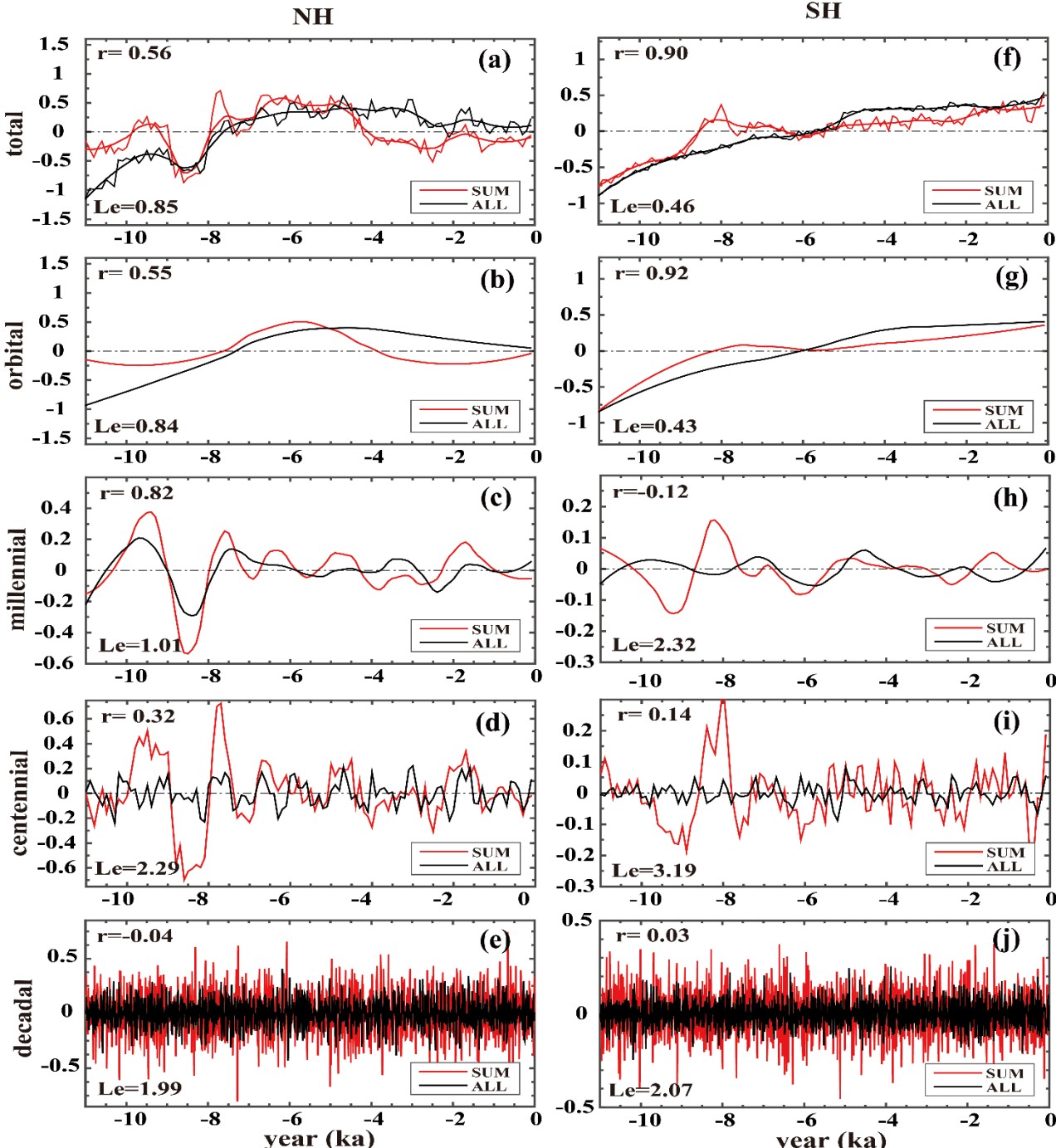

**Figure 2: The surface temperature time series derived from the ALL run (black) and the SUM (red). The (a-e) is similar to the Fig.1a-e but for NH, and the (f-j) is similar to the Fig1.a-e but for SH. The x axis is year (ka, 0 is AD 1950, negative is before AD 1950), and the y axis is temperature anomaly (°C, relative to 11ka-0ka).**

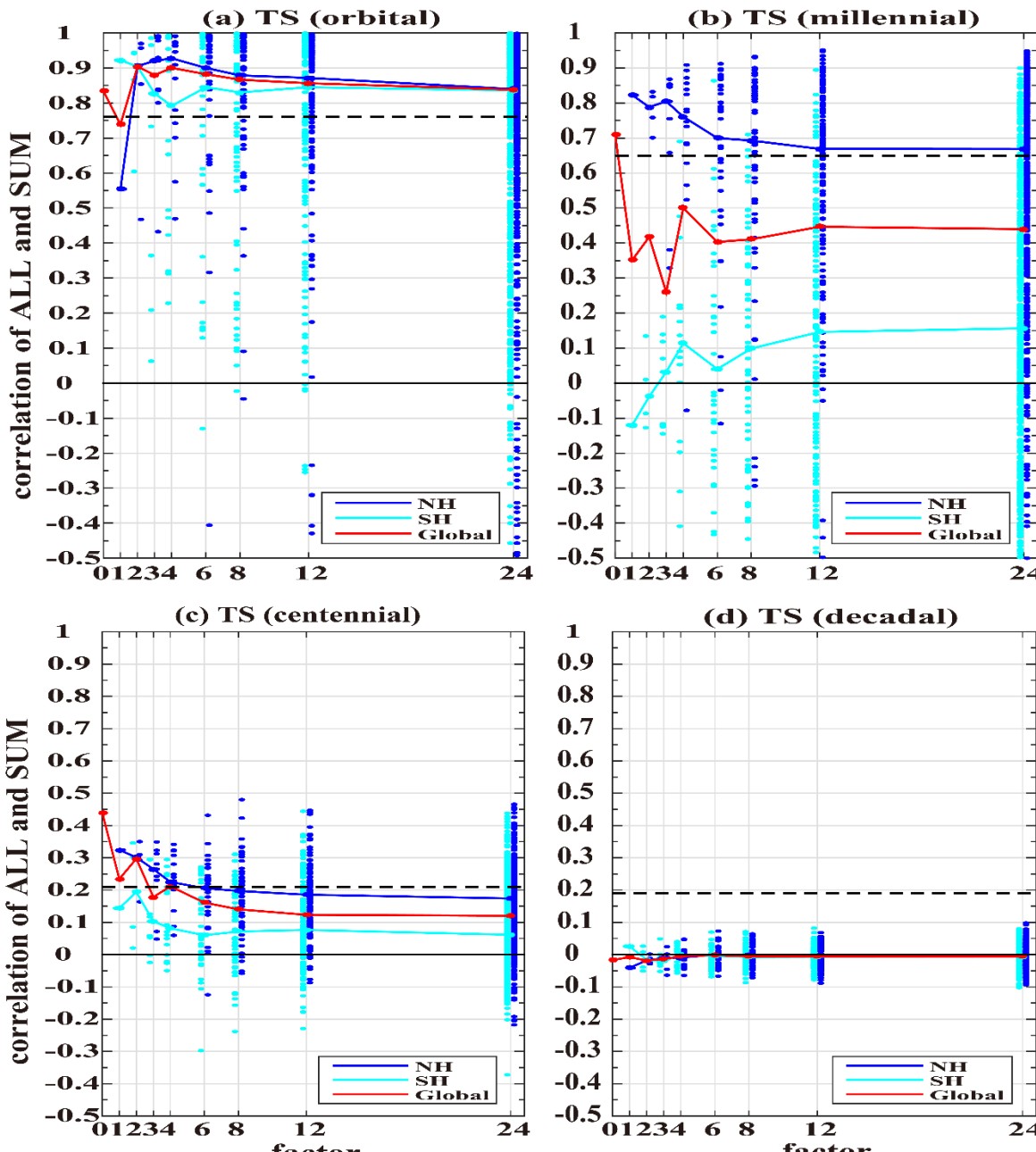

**Figure 3: Correlation coefficient of mean surface temperature between the ALL and SUM outputs in different spatial-time scales. a, orbital; b, millennial; c, centennial; d, decadal. The blue dots for NH, cyan dots for SH and red dots for all the correlation coefficient median in the same spatial scale (i.e. factor) of global. The red line is connecting the median dots at all division factors. The blue (cyan) line is connecting the median of all blue (cyan) dots at all division factors. The black thick dashed line is 95% confidence level. The black thin solid line is zero. The x axis is division factor, and y axis is correlation coefficient. There are only about 10 points with the correlation coefficient lower than -0.5 so they are neglected in a-c.**

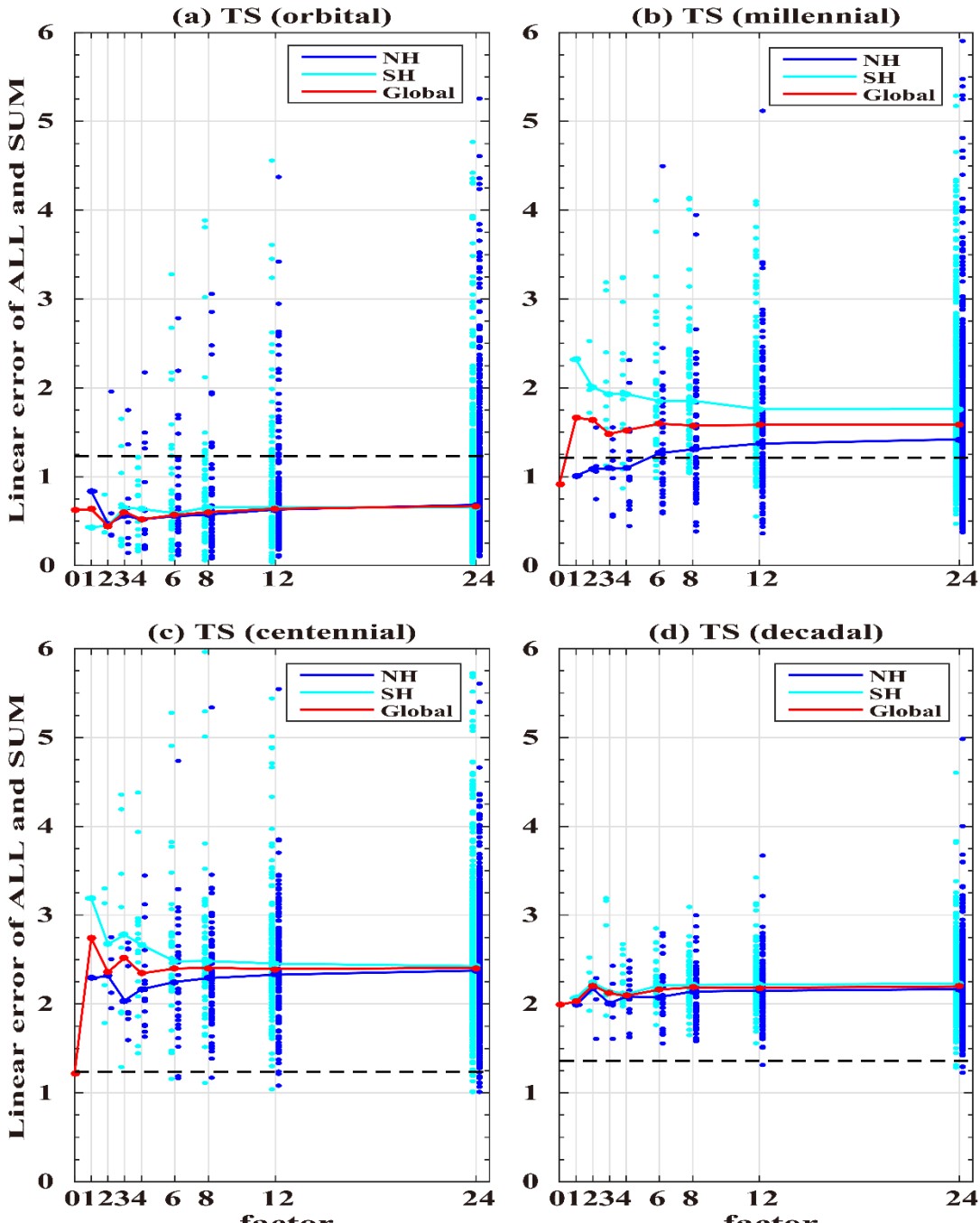

**Figure 4: Same as Figure 3, but for linear error ($L_e$, y axis). The black thick dashed line is 95% confidence level of $L_e$. There are only about 10 points with the $L_e$ larger than 6 so they are neglected in a-c.**

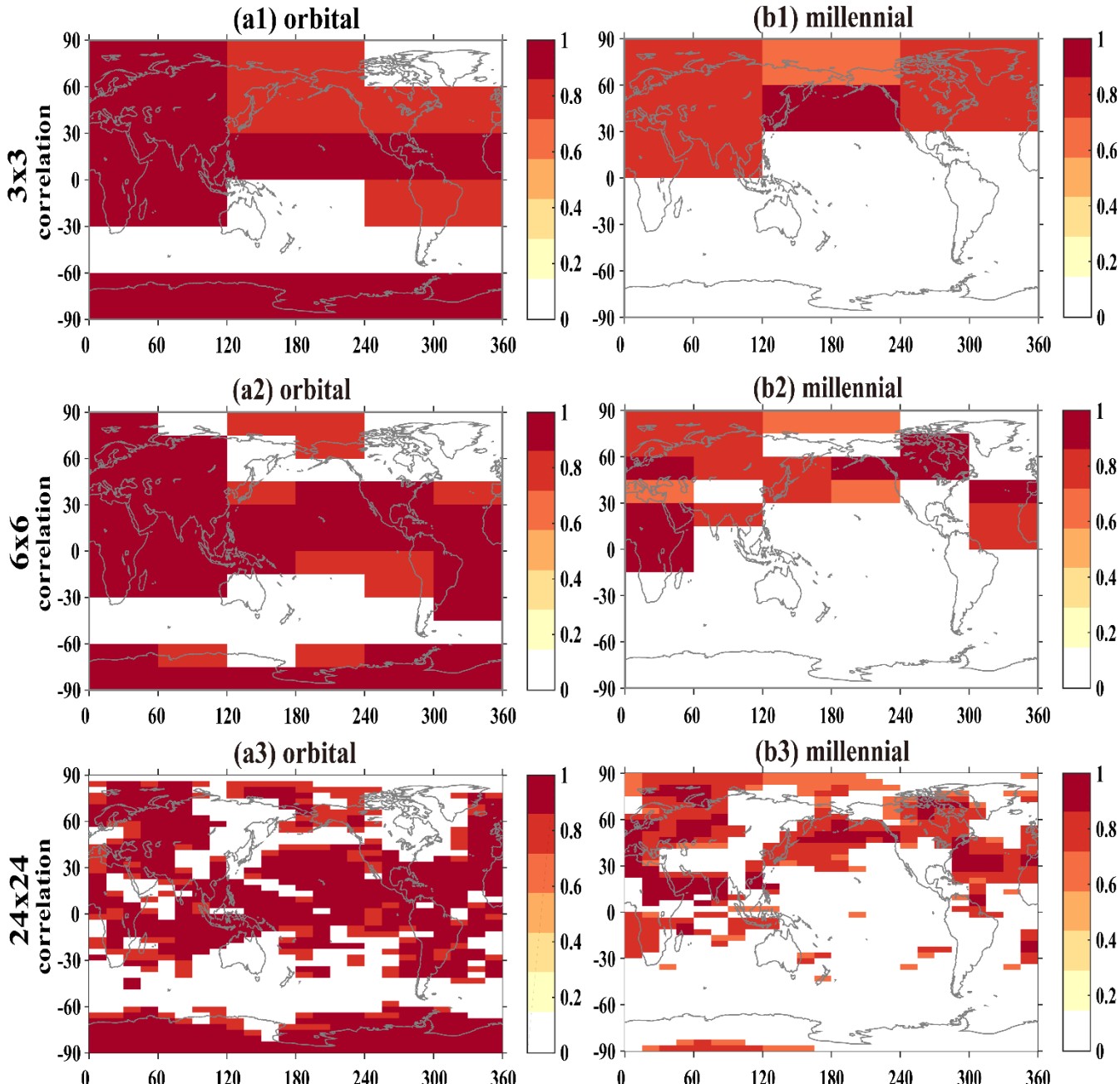

Figure 5: Correlation coefficient of mean surface temperature between the ALL and SUM outputs of the two timescales (a1-a3, orbital; b1-b3, millennial) for three representative spatial scales, *f=3, 6* and *24* (the other factors are not shown because they are similar to the abovementioned three representative spatial scales). Only those regions of significant at 95% confidence level are shaded in colors.

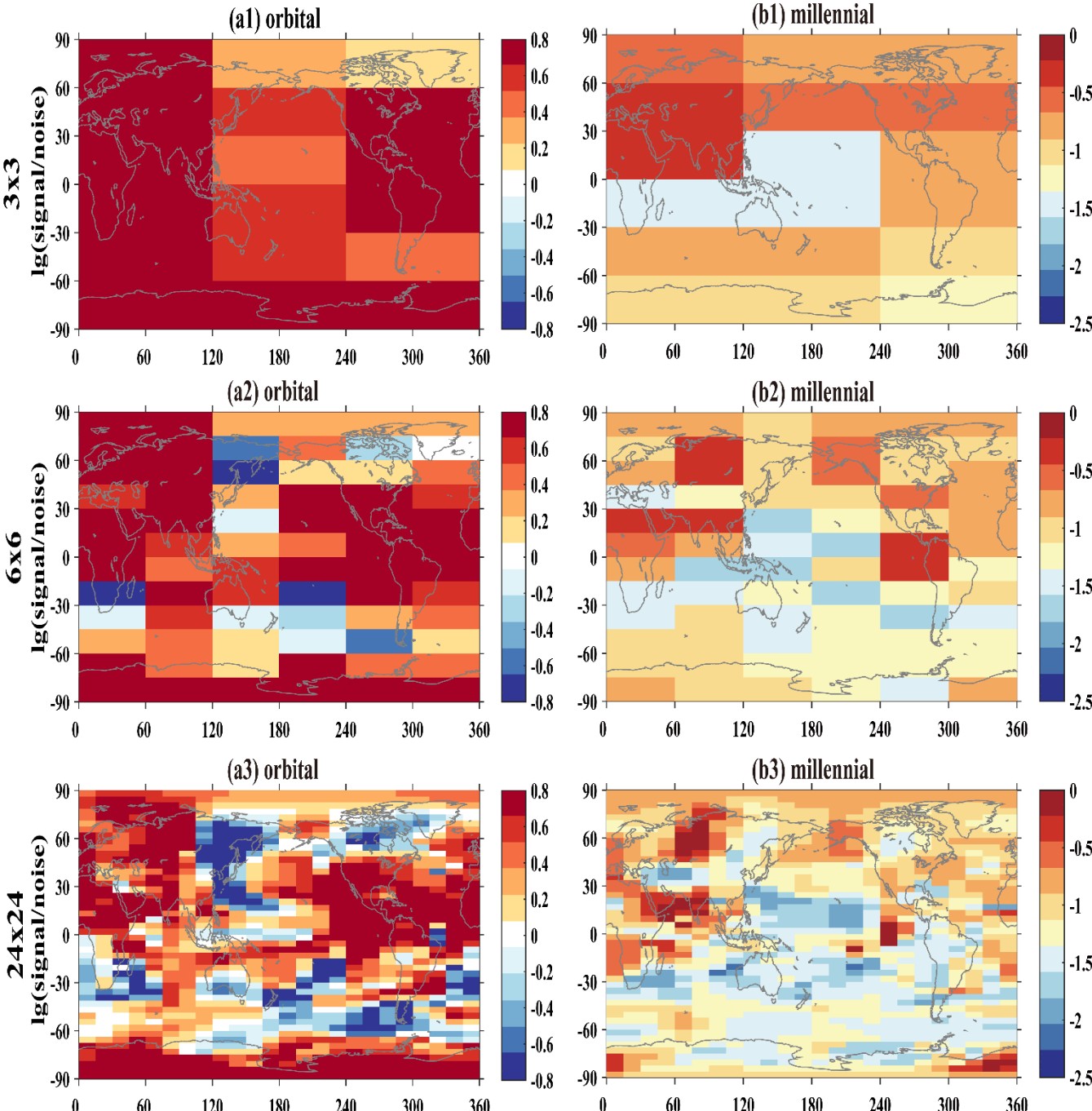

**Figure 6: The signal-to-noise ratios on the orbital (a1-a3) and millennial (b1-b3) timescales derived from the ALL run. Here the signal is used by the orbital (millennial) variability variance, and the noise is used by the sum of the centennial and decadal variabilities variance. The numbers of 1, 2 and 3 are for the three representative spatial scales,** *f=3, 6* **and** *24,* **respectively. In order to show the signal clearly, the log base 10 is taken on the signal to noise ratios.**

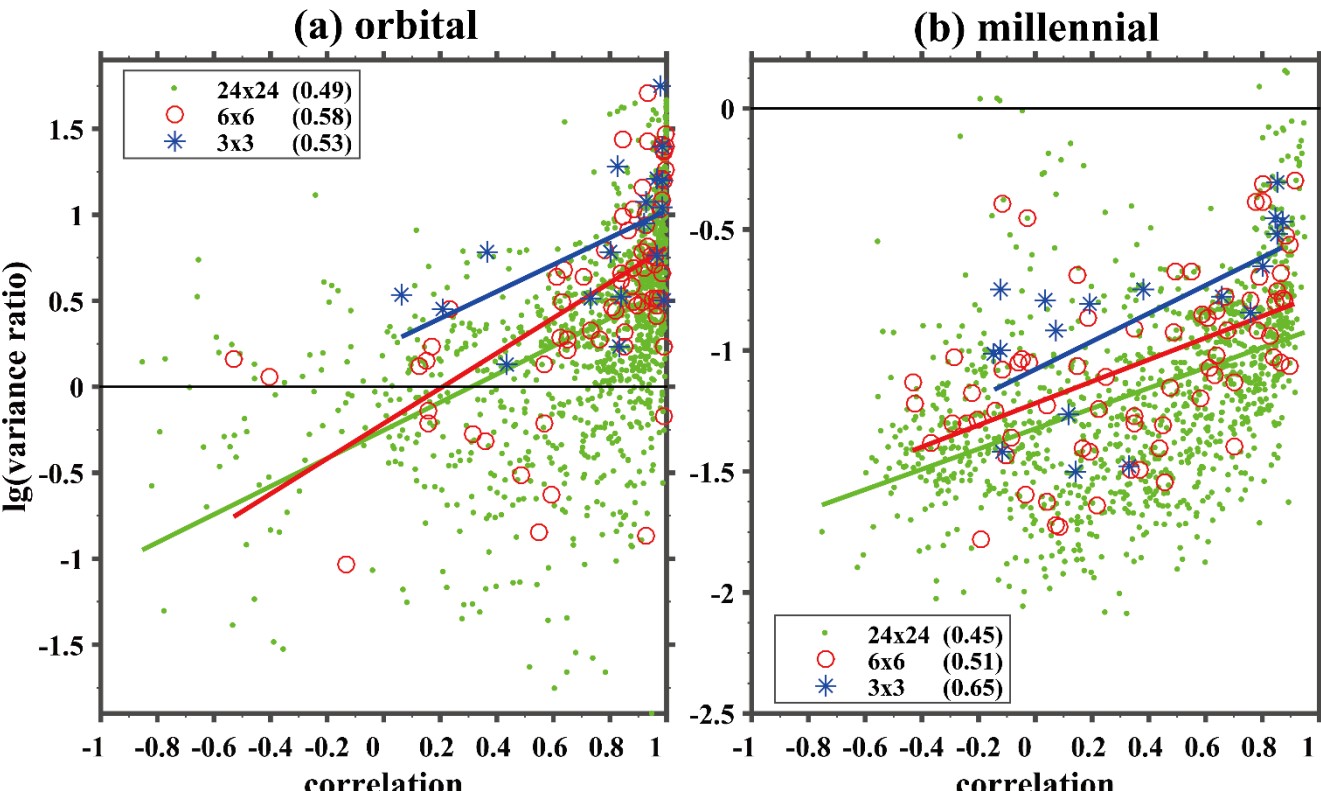

**Figure 7: Scatter diagram of the correlation coefficient and the signal-to-noise ratio (variance ratio) on the orbital (a) and millennial (b) time scale. The log base 10 is taken on the variance ratio, as indicated in Figure 6. Only those for the three representative spatial factors (*f=3, 6* and *24*) are shown in the panels. The blue stars and linear fitting line are for factor 3, the red circles and linear fitting line are for factor 6, the green dots and linear fitting line are for factor 24. The fitting coefficients are list in the brackets at the upper (lower) left corners of the panel a (b).**