# Peer review of "Holocene temperature response to external forcing: Assessing the linear response and its spatial and temporal dependence"

_Climate of the Past, 2018_

## Referee Comment (RC1) · Anonymous Referee #1 · 20 Feb 2019

In this work, the authors make a first attempt to investigate the linearity of forced Holocene variability in dependence on the temporal and spatial scale – by using global surface temperature fields obtained from the TraCE-21ka paleo-climate CGCM simulation. The topic is interesting and the results seem to indicate that further research into this direction may provide further knowledge, that will be useful for both, (a) the interpretation of paleo records and (b) the attribution and detection field of research. Nonetheless, I have a number of concerns regarding the conceptual approach (see 'general comments') which, I think, need clarification before the conclusions, drawn by the authors, can be thoroughly evaluated. A few specific questions are listed at the end (see 'Specific comments').

[Figure]

**1  General comments**

(1) Throughout this work, it seems that the following two *different* questions are mixed up, which makes it basically impossible to evaluate the conclusions drawn from the results:

**Q-1.** How linear is the response to external forcing? If we denote the temperature resonse to the full external forcing, $F_{all}(t) = F_1(t) + F_2(t) + F_3(t) + F_4(t)$, by $T_R(F_{all}(t))$, the response to the individual forcings by $T_R(F_i(t))$ (with $i = 1, \ldots, 4$), and the internal temperature variability of the five model simulations by $T_{I,all}$, $T_{I,1}$, $T_{I,2}$, $T_{I,3}$ and $T_{I,4}$, respectively, then the linearity of the response could be defined by the extent to which the statement

$$T_R(F_{all}(t)) = \sum_{i=1}^{4} T_R(F_i(t)) \tag{1}$$

holds, and the linearity could be measured by the correlation between the forced response on the left and that on the right hand side of the above equation. In the manuscript, however, the correlation is computed (see Section 2.2) from the full 'forced plus internal' temperature variability, i.e. between $T_R(F_{all}(t)) + T_{I,all}(t)$ and $\sum_{i=1}^{4} T_R(F_i(t)) + T_{I,i}(t)$.

Since this latter correlation is influenced by the signal-to-noise ratio, $\mathsf{Var}(T_R)/\mathsf{Var}(T_I)$, a small correlation does *not* necessarily indicate the absence of linearity, because it could be that simply the signal-to-noise ratio is small, although the response is still perfectly linear. (One would need ensembles of model simulations for each of the five forcing scenarios, and then use the ensemble average in order to suppress the internal variability.) Hence, Q-1 cannot be answered by this approach (without additional information), unless one would always obtain correlations close to unity, which would indicate strong linearity.

**Q-2.** What is the relative importance of externally forced vs. internal variability, assum-
ing the response were linear? To answer this question one could use the correlation computed from the full temperature variability, as done in the manuscript, but one had to assume the linearity which, however, is to be proven by this work, in particular, for different temporal and spatial scales.

Hence, the authors should clarify the above issues, and make explicit which of their results contributes to which one of the above two questions. This will also help to clarify the implications of the conclusions for various research fields.

(2) Even if we had ensembles available for each of the forcing scenarios, it would still be possible to obtain a large correlation coefficient although the response is only weakly linear (i.e., mostly non-linear), if the individual response to, for example, one of the forcings $F_i$ is much larger than the responses to the remaining forcings, because in this case the full temperature variability might still be dominated by the response to the strong forcing (the non-linear interactions might still be relatively small). Thus, one would need to know the strength (e.g., in terms of variance) of the responses to the various individual forcings.

(3) In the Conclusions section it should be mentioned that, even if strong linearity for the given model simulations were proven, then this conclusion is valid only for the given range of forcing amplitudes as non-linearities may appear for stronger forcings.

(4) How is it justified to estimate the variance of the internal variability at orbital and millennial time scales by the full variance at centennial and decadal variability (page 7, last paragraph)? That is, why should we have

$$\text{Var}_{orb,mill}(T_{I,all}) \approx \text{Var}_{cent+dec}(T_{I,all} + T_R(F_{all}))? \tag{2}$$

Even if we assume that $\text{Var}_{cent+dec}(T_R(F_{all}))$ is small compared to $\text{Var}_{cent+dec}(T_{I,all})$, this does not imply anything about the relation between $\text{Var}_{cent+dec}(T_{I,all})$ and $\text{Var}_{orb,mill}(T_{I,all})$. Maybe it could be helpful to investigate the power spectra of the temperature variability under the various forcing scenarios?

**2 Specific comments**

(5) It would be nice if the reasons for showing the linear error index $L_e$ were explicitly discussed, and what the implications of this index are for the linearity. And what is the added value of this index over the correlation coefficient?

(6) Please, be a bit more explicit how the significance levels are computed. For example, how is the AR(1) fit done in case of the correlation, and what is the bootstrap design for the error index?

---

## Referee Comment (RC2) · Oliver Bothe (Referee) · 22 Feb 2019

**General Comments**

Wan et al. use the TraCE-21ka simulations in their manuscript "On the linearity of the temperature response in Holocene: the spatial and temporal dependence" to provide an initial assessment of the linearity of the climate response to various assumed forcings. They study this separately for a number of spatial and temporal scales. The idea behind the manuscript can result in a valuable contribution to our understanding of past and future climate changes, to assessing paleo-simulations, and to studying paleo-observational records. I do not have real major concerns but I think various clarifications and additional discussions are necessary before the manuscript could be

accepted for publication. These clarifications should be re-evaluated by a round of revisions and therefore I, nevertheless, recommend major revisions.

**Specific Comments**

1: Could the authors please discuss more clearly, why they think that the assumptions on potential linearity hold (see also the major comments by anonymous referee 1). This discussion could also include, how the specific setup of the TraCE simulations hampers or supports the approach. It may help to include in these discussions a priori knowledge/references on forced and internal variability across temporal and spatial scales.

2: Similarly, I think it is necessary to discuss, at least shortly, how the simulations implement the various forcings and how this may influence the results.

3: Could the authors please discuss, why correlation coefficients and the linear index are appropriate measures of the linearity of the responses as studied. Could they please also clarify, which information is added by the linear index and how to interpret the index in this context.

4: Is the linearity assumption even valid for time scales where internal model processes are known to dominate. That is, we can be quite certain a priori that the decadal scale will be dominated by internal climate over the last 11k years.

5: Could the authors please stress that their conclusions really only hold for the specific setup of the TraCE simulations used.

6: More generally, I think the manuscript is missing a dedicated and thorough discussion-section.

7: Page 1 Line 25ff (P1L25): I do not think the authors show this causality conclusively.

Anyway, the SNR plots do not show or are even intended to show, according to the manuscript, why linearity is consistent between millennial and orbital scales, but why linearity is strong in these regions. That is, in my understanding, the manuscript does not support this sentence in this form.

8: P2L22ff: Could the authors please be more specific how this answer can benefit our understanding?

9: P3L3: I do not think the results by Shakun and colleagues or Marsicek and colleagues allow statements about how reasonable CCSM3's climate sensitivity is.

10: P3L7ff: I think the last part of this sentence requires a reference.

11: P3L8: I am not sure, I completely understand how the authors perform their binning.

12: I find much of the method description on page 4 unclear. For example: I am not fully clear how the authors produce their various time series.

13: P4L19: Is an AR1 process appropriate or could other models be more appropriate?

14: P4L23: Could the authors please give more details on their bootstrap. From my point of view, this description does not allow for reproduction of the significance tests.

15: a) I understand that the author's interest is only in the linearity of the response. However, considering Figure 1, I think, the manuscript will benefit if the authors also discuss the visual discrepancy between the SUM and the ALL series. A difference of about 1K between both series between 11K BP and 3K BP is a relevant feature. Indeed, even if the response is largely linear according to the correlation, the potentially smaller nonlinearity appears to be more important here.
b) To add on this, I wonder whether the setup of the simulations really allow for the analyses? However, I am not familiar enough with the setup of the TraCE "single" forcing simulations.

[Figure]

16: P5L28: Could the authors discuss this complexity in more detail, please?

17: P6L5: As far as I can see the authors do not discuss the reason, at least not in depth.

18: P6L8: Could the authors please be specific, why this should be treated with caution.

19: Considering the centennial time scale: the authors diagnose that the linear response on centennial scales is poor. However, at least in the correlations in Figure 3, there appear to be many regions where linearity still is of modest importance. That is while I agree with the assessment that there is "no strong" linearity, the authors appear to dismiss linearity on centennial scales to easily on page 6.

20: P6L32: The description appears to exclude the continent of Australia.

21: P7L1: a) Could the authors please discuss later on, why the response over the continents should be different from the oceans on these very long time scales. b) Could they please also discuss what the strong internal variability over southern oceans implies for reconstruction efforts.

22: Could the remaining ice sheets and the last freshwater forcing implementations influence the results generally and specifically the poor linearity over North America?

23: P7L18: It suggests so for this set of simulations.

24: P7L29: The authors write at a number of instances "North America" but the results differ notably within North America if I interpret the visualizations correctly.

25: P7L31ff, P8L11ff, Figure 7: I am not sure whether these parts add anything to the other analyses.

26: P8L6: I appear to be unable to see the poor SNR over southern North America.

27: Figure 1: Are the linear errors really the same in panels (a) and (b)?

**Technical Comments**

T1: If I understand it correctly, the manuscript will receive language editing by Copernicus if it is accepted. Nevertheless, I think it will help further reviews if the authors check the language everywhere for clarity and grammatical correctness.

T2: Some of the Figures (particulary Figures 3 to 6) are not publication ready. While I assume that Copernicus is going to assist the authors in this if the manuscript is accepted, it probably would shorten the time between submission and final publication if the authors improve on the Figure-quality for the next round of reviews already.

T3: Could the authors please check that all Figure captions are correct. I was not sure.

T4: P4L20: I think "valid" is not the correct expression, here.

T5: P8L2: Could the authors please skip the exclamation mark.

T6: Acknowledgements: I think the authors have to acknowledge the repository or the persons which/who produced and provided the data. (I assume this was the Climate Data Gateway at NCAR.)

---

## Referee Comment (RC3) · Anonymous Referee #3 · 26 Feb 2019

The linearity of the externally forced temperature evolution during the Holocene is investigated using climate model simulations forced by the total or by individual external forcing factors. In particular, it is tested whether the total forced Holocene temperature variability is a superposition/sum of the individual externally forced temperature responses. Moreover the linearity of the forced temperature response is tested on different spatial and temporal scales. The addressed topic is interesting and important.

Major comments:

- please revise the method section. Sometimes it is not clear what was done and why it was done. Please see specific comments below.

- the discussion should be more extensive, in particular the limitations of the study

[Figure]

(please see the following remarks)

- only a single simulation for each forcing is available. Therefore, a correct definition of external and internal variability is not possible. The internal variability likely differs between the individual simulations and the internal variability is likely not constant during the individual simulations. By summing up the four individual simulations it is not certain that the internal variability cancels out. Moreover, the internal variability might depend on the time and spatial scale. In addition, the ALL-forcing experiment still includes the internal variability. Please make this more clear in the text and discuss.

- an ensemble of Holocene simulations with that model is not available. Therefore, although incorrect, because the internal variability might depend on the forcing, it might be useful to get an estimate of the internal variability of the different time and spatial scales from a long control simulation with the same model.

- I am wondering if it makes sense to investigate the shorter time and also partly the regional scales if only one ensemble member is available. The signal to noise ratio on the shorter time and regional scales might require a larger ensemble size to make a robust statement? Using a control simulation - please see previous point - an estimate of the signal to noise rate might be possible.

- I am wondering if the following definition is useful: "Since our study above shows that the linear response is largely valid for orbital and millennial variability, but not for centennial and decadal variability, we define the variance of the orbital and millennial variability crudely as the linear signals, while define the variance of the sum of the centennial and decadal variability, which is dominated by internal variability, as the linear noise." Please comment.

- Laepple and Huybers (2014) have shown that "a multiproxy estimate of sea surface temperature variability that is consistent between proxy types and with instrumental estimates but strongly diverges from climate model simulations toward longer timescales. At millennial timescales, model−data discrepancies reach two orders of magnitude in

the tropics, indicating substantial problems with models or proxies". Please discuss the implications in the context of the findings

- please describe the filtering method in more detail. It is not clear to me what kind of polynomial was used for the LOESS. Moreover, it is not clear whether the authors used several iteration to get more 'robust' estimates. More important, what is the influence of the LOESS-filtering method on the result, in particular on the linearity of the response.

- please describe the method - used to compute the significance of the correlation - in more detail. If I understand the authors correctly, an AR1 process is only fitted to the ALL-forcing simulation on the different time scales. The Monte-Carlo method is then used to produce an ensemble (PDF) of fitted curves. Then the correlations between the fitted curves and the ALL forcing run are computed and the 95% confidence level is determined afterwards. If I understood the authors correctly, I am wondering if this method is sufficient. I would think that an AR1 process has to be fitted to the ALL forcing run and the superposition (sum of the response of the four individual simulations). Then two ensembles - one for the ALL forcing and one ensemble for the superposition - have to be computed using the Monte-Carlo method. The correlations between these two ensembles have to be used to determine the confidence level. Please make also more clear why you choose the AR1 as a benchmark and how robust the parameter of the AR1 process is, in particular for the orbital time scale.

- it is not clear to me why the authors did not do a spectral analysis of the runs like e.g. wavelet analysis, power spectrum, cross power spectrum ...

- why was the analysis based on the model grid and not on climate modes using e.g. EOF analysis?

Minor comments:

- please be more precise (whole text): please rewrite sentences like 'the linear response is strong' => the response is almost linear; the response is similar to that of a

linear system

- whole text: I would prefer: forcings => forcing factors

- page 3, line 8-9: Please rewrite the sentence

---

## Author Comment (AC1) · 22 Mar 2019

In this work, the authors make a first attempt to investigate the linearity of forced Holocene variability in dependence on the temporal and spatial scale – by using global surface temperature fields obtained from the TraCE-21ka paleo-climate CGCM simulation. The topic is interesting and the results seem to indicate that further research into this direction may provide further knowledge, that will be useful for both, (a) the interpretation of paleo records and (b) the attribution and detection field of research. Nonetheless, I have a number of concerns regarding the conceptual approach (see 'general comments') which, I think, need clarification before the conclusions, drawn by the authors, can be thoroughly evaluated. A few specific questions are listed at the end (see 'Specific comments').

**1   General comments**

(1) Throughout this work, it seems that the following two *different* questions are mixed up, which makes it basically impossible to evaluate the conclusions drawn from the results:

**Q-1.** How linear is the response to external forcing? If we denote the temperature resonse to the full external forcing, $F_{all}(t) = F_1(t) + F_2(t) + F_3(t) + F_4(t)$, by $T_R(F_{all}(t))$, the response to the individual forcings by $T_R(F_i(t))$ (with $i = 1, \ldots, 4$), and the internal temperature variability of the five model simulations by $T_{I,all}$, $T_{I,1}$, $T_{I,2}$, $T_{I,3}$ and $T_{I,4}$, respectively, then the linearity of the response could be defined by the extent to which the statement

$$T_R(F_{all}(t)) = \sum_{i=1}^{4} T_R(F_i(t)) \tag{1}$$

holds, and the linearity could be measured by the correlation between the forced response on the left and that on the right hand side of the above equation. In the manuscript, however, the correlation is computed (see Section 2.2) from the full 'forced plus internal' temperature variability, i.e. between $T_R\big(F_{all}(t)\big) + T_{I,all}(t)$

and $\sum_{i=1}^{4} T_R\big(F_i(t)\big) + T_{I,i}(t)$, Since this latter correlation is influenced by the signal-to-noise ratio, $Var(T_R)/Var(T_I)$, a small correlation does not necessarily indicate the absence of linearity, because it could be that simply the signal-to-noise ratio is small, although the response is still perfectly linear. (One would need ensembles of model simulations for each of the five forcing scenarios, and then use the ensemble average in order to suppress the internal variability.) Hence, Q-1 cannot be answered by this approach (without additional information), unless one would always obtain

correlations close to unity, which would indicate strong linearity.

**Q-2.** What is the relative importance of externally forced vs. internal variability, assuming the response were linear? To answer this question one could use the correlation computed from the full temperature variability, as done in the manuscript, but one had to assume the linearity which, however, is to be proven by this work, in particular, for different temporal and spatial scales.

Hence, the authors should clarify the above issues, and make explicit which of their results contributes to which one of the above two questions. This will also help to clarify the implications of the conclusions for various research fields.

**Reply: We thank the reviewer for this comment. We agree with the reviewer completely on the definition of linearity. We apologize for our ambiguity in the original manuscript. Our focus is really on the slow evolution of temperature in the Holocene that is of comparable time scale to the forcing factors. We have made a major revision. First, we have rewritten all the sections except for section 3. In the revision, we clarified our single realization approach and its potential issues for assessing the linear response (in addition to clarification of the data and methods). Second, we have changed the title to: "Holocene temperature response to external forcing: Assessing the linear response and its spatial and temporal dependence"**

(2) Even if we had ensembles available for each of the forcing scenarios, it would still be possible to obtain a large correlation coefficient although the response is only weakly linear (i.e., mostly non-linear), if the individual response to, for example, one of the forcings $F_i$ is much larger than the responses to the remaining forcings, because in this case the full temperature variability might still be dominated by the response to the strong forcing (the non-linear interactions might still be relatively small). Thus, one would need to know the strength (e.g., in terms of variance) of the responses to the various individual forcings.

**Reply: This raises an excellent point. The explained variance of each forcing factor is indeed very interesting. We think it deserves special attention. Due to the multiple time scales and the strong regional dependence here, however, a detailed study on the variance of each forced response would require much more analyses than in the current paper; it also tends to mix information with our basic information in the first paper here. Therefore, we will still focus on "if the linear response is valid" and will leave the study on "how much each forcing contribute" to a follow-up paper. As for the potential case of a response dominated by a single forcing, it is reasonable to consider this case as a good linear response to the dominant forcing, because the impact from other forcings are negligible anyway.**

(3) In the Conclusions section it should be mentioned that, even if strong linearity for the given model simulations were proven, then this conclusion is valid only for the given range of forcing amplitudes as non-linearities may appear for stronger forcings.

**Reply: Agreed. Comments are added in section 4. "It should therefore be kept in mind that the assessment could differ for different variables, in different models, for different periods and for different sets of forcing factors." "The assessment will be also different if a different period is assessed, e.g. the last 21,000 years; with a large amplitude of climate forcing, the linear response may degenerate in the 21,000-year period."**

(4) How is it justified to estimate the variance of the internal variability at orbital and millennial time scales by the full variance at centennial and decadal variability (page 7, last paragraph)? That is, why should we have

$$\mathrm{Var}_{orb,mill}(T_{I,all}) \approx \mathrm{Var}_{cent+dec}(T_{I,all} + T_R(F_{all}))? \quad (2)$$

Even if we assume that $\mathrm{Var}_{cent+dec}(T_R(F_{all}))$ is small compared to $\mathrm{Var}_{cent+dec}(T_{I,all})$, this does not imply anything about the relation between $\mathrm{Var}_{cent+dec}(T_{I,all})$ and $\mathrm{Var}_{orb,mill}(T_{I,all})$. Maybe it could be helpful to investigate the power spectra of the temperature variability under the various forcing scenarios?

**Reply: We apologize for the ambiguity here. Again, this is an approximation based on linear thinking. Since our forcing, orbital, ice sheet, meltwater and $CO_2$ are of time scales of millennial or longer (we don't have solar variability and volcanic forcing!), we assume that the variability at centennial and decadal time scales are caused mostly by internal variability. This point is clarified now in the revision in subsection 3.3. Since our forcing factors are on millennial and orbital time scales, and the linear response is also largely valid for orbital and millennial variability, we use the variance of the orbital and millennial variability as a crude estimate of the linear response signal. Since there is no centennial and decadal forcing in our model and the response of centennial and decadal variability are not linear response, we use the variance of the sum of the centennial and decadal variability as a rough estimate for internal variability as the linear noise.**

**2  Specific comments**

(5) It would be nice if the reasons for showing the linear error index Le were explicitly discussed, and what the implications of this index are for the linearity. And what is the added value of this index over the correlation coefficient?

**Reply: The correlation represents the similarity of the ALL and SUM, but can't**

evaluate the absolute magnitude of the two responses. Even if two time series is perfectly correlated, their magnitudes can differ by an arbitrary constant. The linear error is to reflect the magnitude of the relative error between the ALL and SUM. More clarifications are added in the text in section 2.2 on this.

(6) Please, be a bit more explicit how the significance levels are computed. For example, how is the AR(1) fit done in case of the correlation, and what is the bootstrap design for the error index?

**Reply: The bootstrap is greatly expanded in the rewritten section 2.2.**

---

## Author Comment (AC2) · 22 Mar 2019

**General Comments**

Wan et al. use the TraCE-21ka simulations in their manuscript "On the linearity of the temperature response in Holocene: the spatial and temporal dependence" to provide an initial assessment of the linearity of the climate response to various assumed forcings. They study this separately for a number of spatial and temporal scales. The idea behind the manuscript can result in a valuable contribution to our understanding of past and future climate changes, to assessing paleo-simulations, and to studying paleo-observational records. I do not have real major concerns but I think various clarifications and additional discussions are necessary before the manuscript could be accepted for publication. These clarifications should be re-evaluated by a round of revisions and therefore I, nevertheless, recommend major revisions.

**Specific Comments**

1: Could the authors please discuss more clearly, why they think that the assumptions on potential linearity hold (see also the major comments by anonymous referee 1). This discussion could also include, how the specific setup of the TraCE simulations hampers or supports the approach. It may help to include in these discussions a priori knowledge/references on forced and internal variability across temporal and spatial scales.

**Reply: This point has been discussed in much more detail in the revision. Section 1 and section 4 are written. More clarifications are also given now. See the reply to referee #1, general comment (1).**

2: Similarly, I think it is necessary to discuss, at least shortly, how the simulations implement the various forcings and how this may influence the results.

**Reply: Thank you for your comments. We have added substantially more details on the simulation in subsection 2.1. We also added a paragraph here on the experimental design and its usefulness for linear response study. "It should be noted that the linear response can't be assessed if the individual forcing experiments are performed with the forcing superimposed one-by-one. In this approach, the four external forcing is added sequentially, for example, first the ice sheet, second the ice sheet plus orbital forcing, third the ice sheet, orbital and GHGs, and finally, applying all four forcing of ice sheet, orbital, GHGs and**

**melting water. In this experimental design, the full forcing response is by default the response of the sum response after adding the four forcing factors together, and therefore can't be used to test the linear response. It should also be noted that our four individual forcing experiments, although in principle feasible for assessing linear response, are not designed optimally for the study of Holocene climate. This is because, except for the variable forcing, all the other three forcing factors is fixed at the 19ka condition. As such, the mean state is perturbed from the glacial state, not a Holocene state. This may have contributed to some unknown deterioration on the linear response discussed later. Nevertheless, we believe, our major conclusion should hold approximately. This is because, partly, the response is indeed almost linear for orbital and millennial variability as will be shown later."**

3: Could the authors please discuss, why correlation coefficients and the linear index are appropriate measures of the linearity of the responses as studied. Could they please also clarify, which information is added by the linear index and how to interpret the index in this context.

**Reply: see Reply to specific comment 5 to referee #1. The correlation represents the similarity of the ALL and SUM, but can't evaluate the absolute magnitude of the two responses. Even if two time series is perfectly correlated, their magnitudes can differ by an arbitrary constant. The linear error is to reflect the magnitude of the relative error between the ALL and SUM. More clarifications are added in the text in section 2.2 on this. "However, the correlation does not address the magnitude of the response. Even if $S_t$ and $T_t$ has a perfect correlation $r=1$, the two time series can still differ by any constant factor in their magnitudes. Therefore, we will also use the linear error index $L_e$ to evaluate the magnitude of the linear response."**

4: Is the linearity assumption even valid for time scales where internal model processes are known to dominate. That is, we can be quite certain a priori that the decadal scale will be dominated by internal climate over the last 11k years.

**Reply: With our approach of single realization, the linear response assumption will fail if the internal variability is dominant. Then, a large ensemble is needed to suppress internal variability. This is discussed in detail in the reply to referee #1, general comment (1).**

5: Could the authors please stress that their conclusions really only hold for the specific setup of the TraCE simulations used.

**Reply: Thank you for the reminder. Yes, we have been specifically clear about this in a sentence and discussion in section 4. "The result here represents the first such assessment and is carried out for a single variable (surface temperature) in**

**a single model (CCSM3). It should therefore be kept in mind that the assessment could differ for different variables, in different models, for different periods and for different sets of forcing factors. For example,…"**

6: More generally, I think the manuscript is missing a dedicated and thorough discussion-section.

**Reply: Thank you for your comments. The last section has now been rewritten and substantial discussions are added. Section 1 is also written to address many potential ambiguities.**

7: Page 1 Line 25ff (P1L25): I do not think the authors show this causality conclusively. Anyway, the SNR plots do not show or are even intended to show, according to the manuscript, why linearity is consistent between millennial and orbital scales, but why linearity is strong in these regions. That is, in my understanding, the manuscript does not support this sentence in this form.

**Reply: We think the referee is referring to this sentence: "On the millennial scale, the linear response is still strong in the NH over many regions, albeit weaker than on the orbital scale". The reviewer is correct in that the paper is not to show millennial and orbital are consistent, it is only to show the linear response in different regions for each time scale separately. The comparison is only made by the comparison between the two in Fig.5, instead of SNR in Fig.6.   Therefore, we think this statement is valid.**

8: P2L22ff: Could the authors please be more specific how this answer can benefit our understanding?

**Reply: The linear response is the base for attributing the response to each forcing factors. Indeed, if a regional response is far away from a linear response (that is the sum response is far away from the total response), the attribution of forcing factor becomes not very useful, because there will be large amount of total response that can't be attributed to any forcing. Indeed, our original motivation for this work was to understand a specific climate response in a particular region (Europe + North America) at millennial time scale. But, we realized that we didn't even know if this response is a good linear response to the four individual forcing in the first place. The work is then expanded systematically to all the regions and at all timescales.   We have added more explanation on this point in the new section 1: introduction and section 4: summary and discussion.**

9: P3L3: I do not think the results by Shakun and colleagues or Marsicek and colleagues allow statements about how reasonable CCSM3's climate sensitivity is.

**Reply: We have relaxed the sentence to "suggesting a potentially reasonable**

**climate sensitivity in CCSM3, at global and continental scales." We think this is a reasonable statement.**

10: P3L7ff: I think the last part of this sentence requires a reference.

**Reply: Thank you for your comments. The reference is Liu et al., 2014, Figure 2A. (We have changed the citation more specifically as: "Figure 2A of Liu et al., 2014"). From this figure we can see in the Holocene period have small change by $CO_2$ than in the deglacial which warming response is dominated by the response to $CO_2$. Since the ice sheet retreating is also an important warming effect in the deglaciation in LOVECLIM and FAMOUS, we have also added this factor in the sentence "as it removes the deglacial warming response that is dominated by the response to increased $CO_2$ and ice sheet retreat".**

[Figure]

11: P3L8: I am not sure, I completely understand how the authors perform their binning.

**Reply: Thank you for your comments. The binning simply means we use the 100 year mean data as one data point. So, the 11,000 years of data is binned into 110 data points, each representing the average of 100-year. Partly, this binning is to be consistent with Marsicek, et al (2018). We have made some clarification on this and sometimes used "mean" to replace "binning".**

12: I find much of the method description on page 4 unclear. For example: I am not fully clear how the authors produce their various time series.

**Reply: We apologize for the ambiguity. We have rewritten this part. Basically, the Loess fit is a low-pass filter. The filtered data is therefore described as low-pass filtered data. This paragraph of deriving the various time series have been written.**

13: P4L19: Is an AR1 process appropriate or could other models be more appropriate?

**Reply: We use the AR(1), instead of white noise, is because we test against the reduced degree of freedom in the low pass the data.**

14: P4L23: Could the authors please give more details on their bootstrap. From my point of view, this description does not allow for reproduction of the significance tests.

**Reply: Thank you for your comments. Take the 100-yr binned data for the Holocene for example. The ALL run global mean temperature time serial have 110 points of data, each representing a 100-yr bin. For one realization, the order of the data is swapped randomly. Then, the sum is used to compare this realization once to derive one Le. Since the randomly swapped realization is not related to the sum response, one should expect a large error Le. Here, we perform the random realizations for 1,000,000 times. This gives us 1,000,000 values of Le, forming the PDF of Le values. The minimum 5% level is then used as the 95% confidence level. One can reproduce the results using the matlab function bootstrap. An introduction on bootstrap is given in the reference Efron, 1979 or Wikipedia (https://en.wikipedia.org/wiki/Bootstrapping_(statistics)). These detailed explanations are added in the revision.**

15: a) I understand that the author's interest is only in the linearity of the response. However, considering Figure 1, I think, the manuscript will benefit if the authors also discuss the visual discrepancy between the SUM and the ALL series. A difference of about 1K between both series between 11K BP and 3K BP is a relevant feature. Indeed, even if the response is largely linear according to the correlation, the potentially smaller nonlinearity appears to be more important here.
b) To add on this, I wonder whether the setup of the simulations really allow for the analyses? However, I am not familiar enough with the setup of the TraCE "single" forcing simulations.

**Reply: Thanks for this good point. Indeed, the assessment of linear response depends on the time period and is a "global" measure here. It does not exclude the discrepancy at some times when the nonlinearity or internal variability can be large. Some comments on this point has been added in the discussion on Fig.1. "It should be noted that, the goodness of the linear response is based on the entire period and is meant for the response of the time scale to be studied. Therefore, even for a good linear response at long time scales, the sum response may still differ from the total response significant at some particular time. For example, for the orbital scale response in Fig.1b, even though the linear response is good according to the correlation and $L_e$, there is a 1$^o$C difference between the sum and total at 11ka and 3ka. Therefore, for the orbital scale response, the linear response mainly refers to the trend-like slow response comparable with orbital scale, instead of response features of shorter time scales." The dependence of the linear response assumption on time period is also discussed in**

**section 4.**

**More descriptions of the TraCE-21ka simulation setup have been added. The relevance for the assessment of linear response has been discussed in the reply to question 2.**

16: P5L28: Could the authors discuss this complexity in more detail, please?

**Reply: Thank you for your comments. What we mean is that the goodness of linear response depends on the region and time scale. This sentence has been changed to: "This suggests that the goodness of the linear response depends on both the region and time scale. This further highlight the need to study the linear response at regional scales."**

17: P6L5: As far as I can see the authors do not discuss the reason, at least not in depth.

**Reply: Thank you for your comments. The reason is discussed in section 3.3. For orbital variability (Fig.5a1-a3), the linear response is strong in most regions in the NH across all three spatial scales, with the correlation coefficients above 0.8. In the SH, the linear response is also strong over the continents, but is poor over the ocean. This leads to the significantly reduced linear response in the SH as discussed in Fig.3a-4a. Since there are more continents in NH than SH and generally speaking, the linear response in continents strong than oceans, the linear response in the NH is better than SH. This is only an explanation from one angle, certainly not a full explanation. We have modified this sentence to: "Part of the reason of the stronger linear response in the NH than over SH will be discussed later"**

18: P6L8: Could the authors please be specific, why this should be treated with caution.

**Reply: Usually, the linear response becomes better for larger spatial scale, because the large spatial average suppresses noise (internal variability). This case is the opposite. A note is added on this.**

19: Considering the centennial time scale: the authors diagnose that the linear response on centennial scales is poor. However, at least in the correlations in Figure 3, there appear to be many regions where linearity still is of modest importance. That is while I agree with the assessment that there is "no strong" linearity, the authors appear to dismiss linearity on centennial scales to easily on page 6.

**Reply: Our statement of "no strong" linearity is derived the statement on the centennial variability: "The median linear response on the centennial timescale**

**in either hemisphere across spatial scales (*f>3*, Fig.3c and Fig.4c) is no longer significant, with few correlation coefficients larger than 0.3 and contributing less than 10% of the variance." We are not very clear what the referee is inferring here.**

20: P6L32: The description appears to exclude the continent of Australia.

**Reply: Thank you for the careful observation. A note is added on Australia.**

21: P7L1: a) Could the authors please discuss later on, why the response over the continents should be different from the oceans on these very long time scales. b) Could they please also discuss what the strong internal variability over southern oceans implies for reconstruction efforts.

**Reply: Good questions! We don't know the reason. A comment is added. We plan to further explore this in the future.**

22: Could the remaining ice sheets and the last freshwater forcing implementations influence the results generally and specifically the poor linearity over North America?

**Reply: Again, this is a good question. We plan to further explore the physical mechanism of the linearity response in the future.**

23: P7L18: It suggests so for this set of simulations.

**Reply: We added "in this model". A general note on the dependence of our results to model, time period, climate variable, et al, is also added in section 4.**

24: P7L29: The authors write at a number of instances "North America" but the results differ notably within North America if I interpret the visualizations correctly.

**Reply: Thank you for the careful observation. We have changed North America to Canada.**

25: P7L31ff, P8L11ff, Figure 7: I am not sure whether these parts add anything to the other analyses.

**Reply: This figure is meant to give some intuition of the scatter of the relationship between correlation and SNR.**

26: P8L6: I appear to be unable to see the poor SNR over southern North America.

**Reply: We clarified it now as "the North America continent outside the central North America".**

27: Figure 1: Are the linear errors really the same in panels (a) and (b)?

**Reply: Thank you for your comments. Yes, they are same. I have check it. The Le of Fig.1a is 0.632 and Fig.1b is 0.626. So they have different in the third decimal. But in this paper we only keep two decimal.**

**Technical Comments**

T1: If I understand it correctly, the manuscript will receive language editing by Copernicus if it is accepted. Nevertheless, I think it will help further reviews if the authors check the language everywhere for clarity and grammatical correctness.

**Reply: Thank you for your comments. We have gone through the manuscript carefully several times.**

T2: Some of the Figures (particulary Figures 3 to 6) are not publication ready. While I assume that Copernicus is going to assist the authors in this if the manuscript is accepted, it probably would shorten the time between submission and final publication if the authors improve on the Figure-quality for the next round of reviews already.

**Reply: We have improve the figures.**

T3: Could the authors please check that all Figure captions are correct. I was not sure.

**Reply: We have checked it again.**

T4: P4L20: I think "valid" is not the correct expression, here.

**Reply: We didn't find valid in P4L20, but in P7L20. We have change the wording.**

T5: P8L2: Could the authors please skip the exclamation mark.

**Reply: Thank you for your comments. All right, fixed it.**

T6: Acknowledgements: I think the authors have to acknowledge the repository or the persons which/who produced and provided the data. (I assume this was the Climate Data Gateway at NCAR.)

**Reply: Thank you for your comments. All right, fixed it.**

---

## Author Comment (AC3) · 22 Mar 2019

The linearity of the externally forced temperature evolution during the Holocene is investigated using climate model simulations forced by the total or by individual external forcing factors. In particular, it is tested whether the total forced Holocene temperature variability is a superposition/sum of the individual externally forced temperature responses. Moreover the linearity of the forced temperature response is tested on different spatial and temporal scales. The addressed topic is interesting and important.

Major comments:

- please revise the method section. Sometimes it is not clear what was done and why it was done. Please see specific comments below.

**Reply: Thank you for your comments. We have revised the method section substantially, adding much more details and clarifications on the model setup, data processing, bootstrap method, et al. see replies to referee #1 and #2.**

- the discussion should be more extensive, in particular the limitations of the study (please see the following remarks)

**Reply: Thank you for your comments. The final section has been written, with much more complete discussion on the limitation of the study here.**

- only a single simulation for each forcing is available. Therefore, a correct definition of external and internal variability is not possible. The internal variability likely differs between the individual simulations and the internal variability is likely not constant during the individual simulations. By summing up the four individual simulations it is not certain that the internal variability cancels out. Moreover, the internal variability might depend on the time and spatial scale. In addition, the ALL-forcing experiment still includes the internal variability. Please make this more clear in the text and discuss.

- an ensemble of Holocene simulations with that model is not available. Therefore, although incorrect, because the internal variability might depend on the forcing, it might be useful to get an estimate of the internal variability of the different time and spatial scales from a long control simulation with the same model.

**Reply: Yes. Section 1 and 4 have been written to clarify this issue. Also, see reply to reviewer #1 on the general questions.**

- I am wondering if it makes sense to investigate the shorter time and also partly the regional scales if only one ensemble member is available. The signal to noise ratio on the shorter time and regional scales might require a larger ensemble size to make a robust statement? Using a control simulation - please see previous point - an estimate of the signal to noise rate might be possible.

**Reply: Agreed. This is only a rough estimation. See the revised section 1 and 4 on the limitations.**

- I am wondering if the following definition is useful: "Since our study above shows that the linear response is largely valid for orbital and millennial variability, but not for centennial and decadal variability, we define the variance of the orbital and millennial variability crudely as the linear signals, while define the variance of the sum of the centennial and decadal variability, which is dominated by internal variability, as the linear noise." Please comment.

**Reply: Given the single realization we have, there is no precise way of separating signal and noise. In this particular case, since all the four forcing factors are at orbital and millennial time scales, the forced signal should be in these long time scales, and the noise should be at shorter time scales, if linear response is assumed (which is largely confirmed). So, this gives a rational to for our crude estimation of signal and noise. If, for example, we discuss volcanic forcing and solar variability, this separation of signal and noise is no longer effect and an ensemble is necessary. This has been discussed now in the revised section 1 and 4.**

- Laepple and Huybers (2014) have shown that "a multiproxy estimate of sea surface temperature variability that is consistent between proxy types and with instrumental estimates but strongly diverges from climate model simulations toward longer timescales. At millennial timescales, model-data discrepancies reach two orders of magnitude in the tropics, indicating substantial problems with models or proxies". Please discuss the implications in the context of the findings

**Reply: A good comment. Our conclusion is valid only for this model. If the model internal variability is indeed so much lower than in the observation, the implication of this study to the real world will be limited. This point is added now in section 4. It is an interesting issue to be explored in the future.**

- please describe the filtering method in more detail. It is not clear to me what kind of polynomial was used for the LOESS. Moreover, it is not clear whether the authors used several iteration to get more 'robust' estimates. More important, what is the influence of the LOESS-filtering method on the result, in particular on the linearity of the response.

**Reply: The filtering is discussed in more detail now. LOESS is used here only as one low pass filter. We use this to be consistent with Marsicek et al (2018). (Our original motivation is to interpret the millennial variability found in Marsicek et al). Marsicek et al also verified the locally weighted regression (Loess) by generalized additive model (GAMM) fit. We test**

**some of our results simply using running mean and the results remain qualitatively similar.**

- please describe the method - used to compute the significance of the correlation – in more detail. If I understand the authors correctly, an AR1 process is only fitted to the ALL-forcing simulation on the different time scales. The Monte-Carlo method is then used to produce an ensemble (PDF) of fitted curves. Then the correlations between the fitted curves and the ALL forcing run are computed and the 95% confidence level is determined afterwards. If I understood the authors correctly, I am wondering if this method is sufficient. I would think that an AR1 process has to be fitted to the ALL forcing run and the superposition (sum of the response of the four individual simulations). Then two ensembles - one for the ALL forcing and one ensemble for the superposition – have to be computed using the Monte-Carlo method. The correlations between these two ensembles have to be used to determine the confidence level. Please make also more clear why you choose the AR1 as a benchmark and how robust the parameter of the AR1 process is, in particular for the orbital time scale.

**Reply: We think that the randomization on ALL should be sufficient. This is because the key here is to use randomization to destroy the serial relation between ALL and sum. This can be done by randomize either ALL or sum, or both of them. Indeed, we have tested both cases, randomizing one or both time series and confirmed they are the same.**

**The reviewer is correct in that, strictly speaking, the AR(1) coefficient should be different for each region and should be used for the test of significance. Here, we used the global mean as a common test, mainly for simplicity. Most importantly, our focus here is on the linear response features over the globe, between different regions. Therefore, a common test makes it easy for comparison among different spatial scales and regions. For example, if regional tests are performed, it will be impossible to plot the significance test on the summery figure of Fig.3 and Fig.4 for comparison of different spatial scales. Similarly, it will be hard to compare the value as well as the significance among different regions and spatial scales in Fig.5 and 6. In addition, the global mean AR(1) is meant as a crude representation of most AR(1)'s for different regions. Indeed, except for the orbital scale, the global mean AR(1) is larger than most of the regional AR(1) so that the global mean AR(1) serves as a stricter test. At the orbital scale, the global mean AR(1) is about the middle of the regional AR(1)'s. Finally, we did emphasize that, if one's focus is on a specific region, the regional AR(1) should be used for re-evaluation of the significance. These points are now discussed explicitly in section 2.2.**

- it is not clear to me why the authors did not do a spectral analysis of the runs like e.g. wavelet analysis, power spectrum, cross power spectrum ...

**Reply: Our study is a first preliminary study. Our interests here is mainly on the linear responses on slow time evolution at the orbital and millennial scales. Given only 11,000 years, it is difficult to derive spectral details with high significance. Nevertheless, we agree it will be interesting to explore the spectral features in the future.**

- why was the analysis based on the model grid and not on climate modes using e.g. EOF analysis?

**Reply: Fixed region is more practical for using model to interpret the real world proxy. Our original motivation is to interpret the regional climate response over North America and Europe as discussed in Marsicek et al (2018).   For overall climate response in the model, it is a good idea to perform this in the EOF space.**

Minor comments:

- please be more precise (whole text): please rewrite sentences like 'the linear response is strong' => the response is almost linear; the response is similar to that of a linear system

**Reply: Thank you for your comments. We have attempted to clarify these terminologies.**

- whole text: I would prefer: forcings => forcing factors
**Reply: Done!**

- page 3, line 8-9: Please rewrite the sentence

**Reply: We have deleted it here and explained the data processing in much more details later in 2.2 as follows:**
 **"To the time scale, we decompose a full 11,000-yr annual temperature time series (from 11 ka to 0 ka) in 100-yr bins (a total of 110 data bins, or points, each representing a 100-yr mean) into three components. The three components are to represent the variability of, roughly, orbital, millennial and centennial timescales. Following Marsicek et al. (2018), we derive the orbital and millennial variability using a low-pass filter called the locally weighted regression fits (Loess fits) (Cleveland, 1979). First, the orbital variability is derived by applying a 6500-yr Loess fit low-pass filter onto the temperature time series, and therefore contains the trend and the slow evolution longer than ~6500 years. Second, we apply a 2500-yr Loess fit low-pass filter onto the temperature time series; then, we derive the millennial variability using this 2500-yr low-pass data subtracting the 6500-yr low-pass data. Finally, centennial variability is derived as the difference between the 100-yr binned temperature time series and the 2500-yr low-pass time series. In addition, we also derive a decadal variability time series. First, we compile the 10-yr bin time series from the original 11,000-yr annual time series (of a total of 1,100 data points, each representing a 10-yr mean). Second, we apply a 100-yr running mean low-pass filter on the time series of the 10-yr binned data. Finally, decadal variability is derived by using the 10-yr binned time series minus its 100-yr running mean time series."**

---

## Author Response (AR3)

**Dear editors and reviewers,**

We have made a major revision on the manuscript cp-2018-177. We have completely **rewritten sections 1, 2 and 4**. There are also **changes on section 3 and figures (in Figs.6, 7)** for better illustration. We also changed the

5      **title** of the paper to: "Holocene temperature response to external forcing: Assessing the linear response and its spatial and temporal dependence".

We also changed the authors' **affiliations** and **corresponding authors** to:
"

Lingfeng Wan[1, 2], Zhengyu Liu[3, 4], Jian Liu[1, 2, 4], Weiyi Sun[1, 2], Bin Liu[1, 2]

10     [1]Key Laboratory for Virtual Geographic Environment, Ministry of Education; State Key Laboratory Cultivation Base of Geographical Environment Evolution of Jiangsu Province; Jiangsu Center for Collaborative Innovation in Geographical Information Resource Development and Application; School of Geography Science, Nanjing Normal University, Nanjing, 210023, China.
[2]Jiangsu Provincial Key Laboratory for Numerical Simulation of Large Scale Complex Systems, School of
15     Mathematical Science, Nanjing Normal University, Nanjing, 210023, China.
[3]Atmospheric Science Program, Department of Geography, Ohio State University, Columbus, OH43210, USA.
[4]Open Studio for the Simulation of Ocean-Climate-Isotope, Qingdao National Laboratory for Marine Science and Technology, Qingdao, 266237, China.

20     *Correspondence*: Jian Liu (njdllj@126.com); Zhengyu Liu (liu.7022@osu.edu)
"

We also checked and added the **references**.

The **point-by-point reply** to all the three reviews are finished.

We hope our revision is satisfactory to the journal.

25     Thanks!

**Best regards!**
**All authors of manuscript cp-2018-177**

**Reply to the Anonymous Referee #1:**
In this work, the authors make a first attempt to investigate the linearity of forced Holocene variability in dependence on the temporal and spatial scale – by using global surface temperature fields obtained from the TraCE-21ka paleo-climate CGCM simulation. The topic is interesting and the results seem to indicate that further research into this direction may provide further knowledge, that will be useful for both, (a) the interpretation of paleo records and (b) the attribution and detection field of research. Nonetheless, I have a number of concerns regarding the conceptual approach (see 'general comments') which, I think, need clarification before the conclusions, drawn by the authors, can be thoroughly evaluated. A few specific questions are listed at the end (see 'Specific comments').

Reply: Thank you for the comments. We have modified the manuscript carefully according to your valuable comments.

**1 General comments**

(1) Throughout this work, it seems that the following two *different* questions are mixed up, which makes it basically impossible to evaluate the conclusions drawn from the results:

**Q-1.** How linear is the response to external forcing? If we denote the temperature resonse to the full external forcing, $F_{all}(t) = F_1(t) + F_2(t) + F_3(t) + F_4(t)$, by $T_R(F_{all}(t))$, the response to the individual forcings by $T_R(F_i(t))$ (with $i = 1, \ldots, 4$), and the internal temperature variability of the five model simulations by $T_{I,all}, T_{I,1}, T_{I,2}, T_{I,3}, T_{I,4}$, respectively, then the linearity of the response could be defined by the extent to which the statement

$$T_R(F_{all}(t)) = \sum_{i=1}^{4} T_R(F_i(t)) \tag{1}$$

holds, and the linearity could be measured by the correlation between the forced response on the left and that on the right hand side of the above equation. In the manuscript, however, the correlation is computed (see Section 2.2) from the full 'forced plus internal' temperature variability, i.e. between $T_R(F_{all}(t)) + T_{I,all}(t)$ and $\sum_{i=1}^{4} T_R(F_i(t)) + T_{I,i}(t)$, Since this latter correlation is influenced by the signal-to-noise ratio, $\text{Var}(T_R)/\text{Var}(T_I)$, a small correlation does not necessarily indicate the absence of linearity, because it could be that simply the signal-to-noise ratio is small, although the response is still perfectly linear. (One would need ensembles of model simulations for each of the five forcing scenarios, and then use

the ensemble average in order to suppress the internal variability.) Hence, Q-1 cannot be answered by this approach (without additional information), unless one would always obtain correlations close to unity, which would indicate strong linearity.

**Q-2.** What is the relative importance of externally forced vs. internal variability, assuming the response were linear? To answer this question one could use the correlation computed from the full temperature variability, as done in the manuscript, but one had to assume the linearity which, however, is to be proven by this work, in particular, for different temporal and spatial scales.

Hence, the authors should clarify the above issues, and make explicit which of their results contributes to which one of the above two questions. This will also help to clarify the implications of the conclusions for various research fields.

Reply: We thank the reviewer for this comment. We agree with the reviewer completely on the definition of linearity. We apologize for our ambiguity in the original manuscript. Our focus is really on the slow evolution of temperature in the Holocene that is of comparable time scale to the forcing factors. We have made a major revision. First, we have rewritten all the sections except for section 3. In the revision, we clarified our single realization approach and its potential issues for assessing the linear response (in addition to clarification of the data and methods). Second, we have changed the title to: "Holocene temperature response to external forcing: Assessing the linear response and its spatial and temporal dependence"

(2) Even if we had ensembles available for each of the forcing scenarios, it would still be possible to obtain a large correlation coefficient although the response is only weakly linear (i.e., mostly non-linear), if the individual response to, for example, one of the forcings $F_i$ is much larger than the responses to the remaining forcings, because in this case the full temperature variability might still be dominated by the response to the strong forcing (the non-linear interactions might still be relatively small). Thus, one would need to know the strength (e.g., in terms of variance) of the responses to the various individual forcings.

Reply: This raises an excellent point. The explained variance of each forcing factor is indeed very interesting. We think it deserves special attention. Due to the multiple time scales and the strong regional dependence here, however, a detailed study on the variance of each forced response would require much more analyses than in the current paper; it also tends to mix information with our basic information in the first paper here. Therefore, we will still focus on "if the linear response is valid" and will leave the study on "how much each forcing contribute" to a follow-up paper. As for the potential case of a response dominated by a single forcing, it is reasonable to consider this case as a good linear response to the dominant forcing, because the impact from other forcing factors are negligible anyway.

(3) In the Conclusions section it should be mentioned that, even if strong linearity for the given model simulations were proven, then this conclusion is valid only for the given range of forcing amplitudes as non-linearities may appear for stronger forcings.

Reply: Agreed. Comments are added in section 4. "It should therefore be kept in mind that the assessment could differ for different variables, in different models, for different periods and for different sets of forcing factors." "The assessment will

be also different if a different period is assessed, e.g. the last 21,000 years; with a large amplitude of climate forcing, the linear response may degenerate in the 21,000-year period."

(4) How is it justified to estimate the variance of the internal variability at orbital and millennial time scales by the full variance at centennial and decadal variability (page 7, last paragraph)? That is, why should we have

$$\text{Var}_{orb,mill}(T_{I,all}) \approx \text{Var}_{cent+dec}\left(T_{I,all} + T_R(F_{all})\right)? \qquad (2)$$

Even if we assume that $\text{Var}_{cent+dec}(T_R(F_{all}))$ is small compared to $\text{Var}_{cent+dec}(T_{I,all})$, this does not imply anything about the relation between $\text{Var}_{cent+dec}(T_{I,all})$ and $\text{Var}_{orb,mill}(T_{I,all})$. Maybe it could be helpful to investigate the power spectra of the temperature variability under the various forcing scenarios?

Reply: We apologize for the ambiguity here. Again, this is an approximation based on linear thinking. Since our forcing, orbital, ice sheet, meltwater and $CO_2$ are of time scales of millennial or longer (we don't have solar variability and volcanic forcing!), we assume that the variability at centennial and decadal time scales are caused mostly by internal variability. This point is clarified now in the revision in subsection 3.3. Since our forcing factors are on millennial and orbital time scales, and the linear response is also largely valid for orbital and millennial variability, we use the variance of the orbital and millennial variability as a crude estimate of the linear response signal. Since there is no centennial and decadal forcing in our model and the response of centennial and decadal variability are not linear response, we use the variance of the sum of the centennial and decadal variability as a rough estimate for internal variability as the linear noise.

**2 Specific comments**

(5) It would be nice if the reasons for showing the linear error index Le were explicitly discussed, and what the implications of this index are for the linearity. And what is the added value of this index over the correlation coefficient?

Reply: The correlation represents the similarity of the ALL and SUM, but can't evaluate the absolute magnitude of the two responses. Even if two time series is perfectly correlated, their magnitudes can differ by an arbitrary constant. The linear error is to reflect the magnitude of the relative error between the ALL and SUM. More clarifications are added in the text in section 2.2 on this.

(6) Please, be a bit more explicit how the significance levels are computed. For example, how is the AR(1) fit done in case of the correlation, and what is the bootstrap design for the error index?

Reply: The bootstrap is greatly expanded in the rewritten section 2.2.

**Reply to the Oliver Bothe (Referee #2):**
Wan et al. use the TraCE-21ka simulations in their manuscript "On the linearity of the temperature response in Holocene: the spatial and temporal dependence" to provide an initial assessment of the linearity of the climate response to various assumed forcings. They study this separately for a number of spatial and temporal scales. The idea behind the manuscript can result in a valuable contribution to our understanding of past and future climate changes, to assessing paleo-simulations, and to

15 studying paleo-observational records. I do not have real major concerns but I think various clarifications and additional discussions are necessary before the manuscript could be accepted for publication. These clarifications should be re-evaluated by a round of revisions and therefore I, nevertheless, recommend major revisions.

Reply: Thank you for your comments. We have revised the manuscript according to your valuable comments.

20 **Specific Comments**

1: Could the authors please discuss more clearly, why they think that the assumptions on potential linearity hold (see also the major comments by anonymous referee 1). This discussion could also include, how the specific setup of the TraCE simulations hampers or supports the approach. It may help to include in these discussions a priori knowledge/references on forced and internal variability across temporal and spatial scales.

25 Reply: This point has been discussed in much more detail in the revision. Section 1 and section 4 are written. More clarifications are also given now. See the reply to referee #1, general comment (1).

2: Similarly, I think it is necessary to discuss, at least shortly, how the simulations implement the various forcings and how this may influence the results.

30 Reply: Thank you for your comments. We have added substantially more details on the simulation in subsection 2.1. We also added a paragraph here on the experimental design and its usefulness for linear response study. "It should be noted that the linear response can't be assessed if the individual forcing experiments are performed with the forcing superimposed one-by-one. In this approach, the four external forcing is added sequentially, for example, first the ice sheet, second the ice sheet plus orbital forcing, third the ice sheet, orbital and GHGs, and finally, applying all four forcing of ice sheet, orbital, GHGs

and melting water. In this experimental design, the full forcing response is by default the response of the sum response after adding the four forcing factors together, and therefore can't be used to test the linear response. It should also be noted that our four individual forcing experiments, although in principle feasible for assessing linear response, are not designed optimally for the study of Holocene climate. This is because, except for the variable forcing, all the other three forcing factors is fixed at the 19ka condition. As such, the mean state is perturbed from the glacial state, not a Holocene state. This may have contributed to some unknown deterioration on the linear response discussed later. Nevertheless, we believe, our major conclusion should hold approximately. This is because, partly, the response is indeed almost linear for orbital and millennial variability as will be shown later."

3: Could the authors please discuss, why correlation coefficients and the linear index are appropriate measures of the linearity of the responses as studied. Could they please also clarify, which information is added by the linear index and how to interpret the index in this context.

Reply: see Reply to specific comment 5 to referee #1. The correlation represents the similarity of the ALL and SUM, but can't evaluate the absolute magnitude of the two responses. Even if two time series is perfectly correlated, their magnitudes can differ by an arbitrary constant. The linear error is to reflect the magnitude of the relative error between the ALL and SUM. More clarifications are added in the text in section 2.2 on this. "However, the correlation does not address the magnitude of the response. Even if $S_t$ and $T_t$ has a perfect correlation $r=1$, the two time series can still differ by any constant factor in their magnitudes. Therefore, we will also use the linear error index $L_e$ to evaluate the magnitude of the linear response."

4: Is the linearity assumption even valid for time scales where internal model processes are known to dominate. That is, we can be quite certain a priori that the decadal scale will be dominated by internal climate over the last 11k years.

Reply: With our approach of single realization, the linear response assumption will fail if the internal variability is dominant. Then, a large ensemble is needed to suppress internal variability. This is discussed in detail in the reply to referee #1, general comment (1).

5: Could the authors please stress that their conclusions really only hold for the specific setup of the TraCE simulations used.

Reply: Thank you for the reminder. Yes, we have been specifically clear about this in a sentence and discussion in section 4. "The result here represents the first such assessment and is carried out for a single variable (surface temperature) in a single model (CCSM3). It should therefore be kept in mind that the assessment could differ for different variables, in different models, for different periods and for different sets of forcing factors. For example, ... "

6: More generally, I think the manuscript is missing a dedicated and thorough discussion-section.

Reply: Thank you for your comments. The last section has now been rewritten and substantial discussions are added. Section 1 is also written to address many potential ambiguities.

7: Page 1 Line 25ff (P1L25): I do not think the authors show this causality conclusively. Anyway, the SNR plots do not show or are even intended to show, according to the manuscript, why linearity is consistent between millennial and orbital scales, but why linearity is strong in these regions. That is, in my understanding, the manuscript does not support this sentence in this form.

Reply: We think the referee is referring to this sentence: "On the millennial scale, the linear response is still strong in the NH over many regions, albeit weaker than on the orbital scale". The reviewer is correct in that the paper is not to show millennial and orbital are consistent, it is only to show the linear response in different regions for each time scale separately. The comparison is only made by the comparison between the two in Fig.5, instead of SNR in Fig.6. Therefore, we think this statement is valid.

8: P2L22ff: Could the authors please be more specific how this answer can benefit our understanding?

Reply: The linear response is the base for attributing the response to each forcing factors. Indeed, if a regional response is far away from a linear response (that is the sum response is far away from the total response), the attribution of forcing factor becomes not very useful, because there will be large amount of total response that can't be attributed to any forcing. Indeed, our original motivation for this work was to understand a specific climate response in a particular region (Europe + North America) at millennial time scale. But, we realized that we didn't even know if this response is a good linear response to the four individual forcing in the first place. The work is then expanded systematically to all the regions and at all timescales. We have added more explanation on this point in the new section 1: introduction and section 4: summary and discussion.

9: P3L3: I do not think the results by Shakun and colleagues or Marsicek and colleagues allow statements about how reasonable CCSM3's climate sensitivity is.

Reply: We have relaxed the sentence to "suggesting a potentially reasonable climate sensitivity in CCSM3, at global and continental scales." We think this is a reasonable statement.

10: P3L7ff: I think the last part of this sentence requires a reference.

Reply: Thank you for your comments. The reference is Liu et al., 2014, Figure 2A. (We have changed the citation more specifically as: "Figure 2A of Liu et al., 2014"). From this figure we can see in the Holocene period have small change by $CO_2$ than in the deglacial which warming response is dominated by the response to $CO_2$. Since the ice sheet retreating is also an important warming effect in the deglaciation in LOVECLIM and FAMOUS, we have also added this factor in the sentence "as it removes the deglacial warming response that is dominated by the response to increased $CO_2$ and ice sheet retreat".

[Figure]

11: P3L8: I am not sure, I completely understand how the authors perform their binning.

Reply: Thank you for your comments. The binning simply means we use the 100 year mean data as one data point. So, the 11,000 years of data is binned into 110 data points, each representing the average of 100-year. Partly, this binning is to be consistent with Marsicek, et al (2018). We have made some clarification on this and sometimes used "mean" to replace "binning".

12: I find much of the method description on page 4 unclear. For example: I am not fully clear how the authors produce their various time series.

Reply: We apologize for the ambiguity. We have rewritten this part. Basically, the Loess fit is a low-pass filter. The filtered data is therefore described as low-pass filtered data. This paragraph of deriving the various time series have been written.

13: P4L19: Is an AR1 process appropriate or could other models be more appropriate?

Reply: We use the AR(1), instead of white noise, is because we test against the reduced degree of freedom in the low pass the data.

14: P4L23: Could the authors please give more details on their bootstrap. From my point of view, this description does not allow for reproduction of the significance tests.

Reply: Thank you for your comments. Take the 100-yr binned data for the Holocene for example. The ALL run global mean temperature time serial have 110 points of data, each representing a 100-yr bin. For one realization, the order of the data is swapped randomly. Then, the sum is used to compare this realization once to derive one Le. Since the randomly swapped realization is not related to the sum response, one should expect a large error Le. Here, we perform the random realizations for 1,000,000 times. This gives us 1,000,000 values of Le, forming the PDF of Le values. The minimum 5% level is then used as the 95% confidence level. One can reproduce the results using the matlab function bootstrap. An introduction on bootstrap is given in the reference Efron, 1979 or Wikipedia (https://en.wikipedia.org/wiki/Bootstrapping_(statistics)). These detailed explanations are added in the revision.

15: a) I understand that the author's interest is only in the linearity of the response. However, considering Figure 1, I think, the manuscript will benefit if the authors also discuss the visual discrepancy between the SUM and the ALL series. A difference of about 1K between both series between 11K BP and 3K BP is a relevant feature. Indeed, even if the response is largely linear according to the correlation, the potentially smaller nonlinearity appears to be more important here.

b) To add on this, I wonder whether the setup of the simulations really allow for the analyses? However, I am not familiar enough with the setup of the TraCE "single" forcing simulations.

Reply: Thanks for this good point. Indeed, the assessment of linear response depends on the time period and is a "global" measure here. It does not exclude the discrepancy at some times when the nonlinearity or internal variability can be large. Some comments on this point has been added in the discussion on Fig.1. "It should be noted that, the goodness of the linear response is based on the entire period and is meant for the response of the time scale to be studied. Therefore, even for a good linear response at long time scales, the sum response may still differ from the total response significant at some particular time. For example, for the orbital scale response in Fig.1b, even though the linear response is good according to the correlation and $L_e$, there is a 1°C difference between the sum and total at 11ka and 3ka. Therefore, for the orbital scale response, the linear response mainly refers to the trend-like slow response comparable with orbital scale, instead of response features of shorter time scales." The dependence of the linear response assumption on time period is also discussed in section 4.

More descriptions of the TraCE-21ka simulation setup have been added. The relevance for the assessment of linear response has been discussed in the reply to question 2.

16: P5L28: Could the authors discuss this complexity in more detail, please?

Reply: Thank you for your comments. What we mean is that the goodness of linear response depends on the region and time scale. This sentence has been changed to: "This suggests that the goodness of the linear response depends on both the region and time scale. This further highlight the need to study the linear response at regional scales."

17: P6L5: As far as I can see the authors do not discuss the reason, at least not in depth.

Reply: Thank you for your comments. The reason is discussed in section 3.3. For orbital variability (Fig.5a1-a3), the linear response is strong in most regions in the NH across all three spatial scales, with the correlation coefficients above 0.8. In the SH, the linear response is also strong over the continents, but is poor over the ocean. This leads to the significantly reduced linear response in the SH as discussed in Fig.3a-4a. Since there are more continents in NH than SH and generally speaking, the linear response in continents strong than oceans, the linear response in the NH is better than SH. This is only an explanation from one angle, certainly not a full explanation. We have modified this sentence to: "Part of the reason of the stronger linear response in the NH than over SH will be discussed later"

18: P6L8: Could the authors please be specific, why this should be treated with caution.

Reply: Usually, the linear response becomes better for larger spatial scale, because the large spatial average suppresses noise (internal variability). This case is the opposite. A note is added on this.

19: Considering the centennial time scale: the authors diagnose that the linear response on centennial scales is poor. However, at least in the correlations in Figure 3, there appear to be many regions where linearity still is of modest importance. That is while I agree with the assessment that there is "no strong" linearity, the authors appear to dismiss linearity on centennial scales to easily on page 6.

Reply: Our statement of "no strong" linearity is derived the statement on the centennial variability: "The median linear response on the centennial timescale in either hemisphere across spatial scales ($f>3$, Fig.3c and Fig.4c) is no longer significant, with few correlation coefficients larger than 0.3 and contributing less than 10% of the variance." We are not very clear what the referee is inferring here.

20: P6L32: The description appears to exclude the continent of Australia.

Reply: Thank you for the careful observation. A note is added on Australia.

21: P7L1: a) Could the authors please discuss later on, why the response over the continents should be different from the oceans on these very long time scales. b) Could they please also discuss what the strong internal variability over southern oceans implies for reconstruction efforts.

Reply: Good questions! We don't know the reason. A comment is added. We plan to further explore this in the future.

22: Could the remaining ice sheets and the last freshwater forcing implementations influence the results generally and specifically the poor linearity over North America?

Reply: Again, this is a good question. We plan to further explore the physical mechanism of the linearity response in the future.

23: P7L18: It suggests so for this set of simulations.

Reply: We added "in this model". A general note on the dependence of our results to model, time period, climate variable, et al, is also added in section 4.

24: P7L29: The authors write at a number of instances "North America" but the results differ notably within North America if I interpret the visualizations correctly.

Reply: Thank you for the careful observation. We have changed North America to Canada.

25: P7L31ff, P8L11ff, Figure 7: I am not sure whether these parts add anything to the other analyses.

Reply: This figure is meant to give some intuition of the scatter of the relationship between correlation and SNR.

26: P8L6: I appear to be unable to see the poor SNR over southern North America.

Reply: We clarified it now as "the North America continent outside the central North America".

27: Figure 1: Are the linear errors really the same in panels (a) and (b)?

Reply: Thank you for your comments. Yes, they are same. I have check it. The Le of Fig.1a is 0.632 and Fig.1b is 0.626. So they have different in the third decimal. But in this paper we only keep two decimal.

**Technical Comments**

T1: If I understand it correctly, the manuscript will receive language editing by Copernicus if it is accepted. Nevertheless, I think it will help further reviews if the authors check the language everywhere for clarity and grammatical correctness.

Reply: Thank you for your comments. We have gone through the manuscript carefully several times.

T2: Some of the Figures (particulary Figures 3 to 6) are not publication ready. While I assume that Copernicus is going to assist the authors in this if the manuscript is accepted, it probably would shorten the time between submission and final publication if the authors improve on the Figure-quality for the next round of reviews already.

Reply: We have improve the figures.

T3: Could the authors please check that all Figure captions are correct. I was not sure.

Reply: We have checked it again.

T4: P4L20: I think "valid" is not the correct expression, here.

Reply: We didn't find valid in P4L20, but in P7L20. We have change the wording.

T5: P8L2: Could the authors please skip the exclamation mark.

Reply: Thank you for your comments. All right, fixed it.

T6: Acknowledgements: I think the authors have to acknowledge the repository or the persons which/who produced and provided the data. (I assume this was the Climate Data Gateway at NCAR.)

Reply: Thank you for your comments. All right, fixed it.
The linearity of the externally forced temperature evolution during the Holocene is investigated using climate model
10   simulations forced by the total or by individual external forcing factors. In particular, it is tested whether the total forced Holocene temperature variability is a superposition/sum of the individual externally forced temperature responses. Moreover the linearity of the forced temperature response is tested on different spatial and temporal scales. The addressed topic is interesting and important.

Reply: Thank you for your comments. We have revised the manuscript according to your valuable comments.

Major comments:

- please revise the method section. Sometimes it is not clear what was done and why it was done. Please see specific comments below.

Reply: Thank you for your comments. We have revised the method section substantially, adding much more details and
20   clarifications on the model setup, data processing, bootstrap method, et al. see replies to referee #1 and #2.

- the discussion should be more extensive, in particular the limitations of the study (please see the following remarks)

Reply: Thank you for your comments. The final section has been written, with much more complete discussion on the limitation of the study here.

- only a single simulation for each forcing is available. Therefore, a correct definition of external and internal variability is not possible. The internal variability likely differs between the individual simulations and the internal variability is likely not constant during the individual simulations. By summing up the four individual simulations it is not certain that the internal variability cancels out. Moreover, the internal variability might depend on the time and spatial scale. In addition, the ALL-
30   forcing experiment still includes the internal variability. Please make this more clear in the text and discuss.

- an ensemble of Holocene simulations with that model is not available. Therefore, although incorrect, because the internal variability might depend on the forcing, it might be useful to get an estimate of the internal variability of the different time and spatial scales from a long control simulation with the same model.

Reply: Yes. Section 1 and 4 have been written to clarify this issue. Also, see reply to reviewer #1 on the general questions.

- I am wondering if it makes sense to investigate the shorter time and also partly the regional scales if only one ensemble member is available. The signal to noise ratio on the shorter time and regional scales might require a larger ensemble size to make a robust statement? Using a control simulation - please see previous point - an estimate of the signal to noise rate might be possible.

10    Reply: Agreed. This is only a rough estimation. See the revised section 1 and 4 on the limitations.

- I am wondering if the following definition is useful: "Since our study above shows that the linear response is largely valid for orbital and millennial variability, but not for centennial and decadal variability, we define the variance of the orbital and millennial variability crudely as the linear signals, while define the variance of the sum of the centennial and decadal

15    variability, which is dominated by internal variability, as the linear noise." Please comment.

Reply: Given the single realization we have, there is no precise way of separating signal and noise. In this particular case, since all the four forcing factors are at orbital and millennial time scales, the forced signal should be in these long time scales, and the noise should be at shorter time scales, if linear response is assumed (which is largely confirmed). So, this gives a rational to for our crude estimation of signal and noise. If, for example, we discuss volcanic forcing and solar variability,

20    this separation of signal and noise is no longer effect and an ensemble is necessary. This has been discussed now in the revised section 1 and 4.

- Laepple and Huybers (2014) have shown that "a multiproxy estimate of sea surface temperature variability that is consistent between proxy types and with instrumental estimates but strongly diverges from climate model simulations

25    toward longer timescales. At millennial timescales, model-data discrepancies reach two orders of magnitude in the tropics, indicating substantial problems with models or proxies". Please discuss the implications in the context of the findings

Reply: A good comment. Our conclusion is valid only for this model. If the model internal variability is indeed so much lower than in the observation, the implication of this study to the real world will be limited. This point is added now in section 4. It is an interesting issue to be explored in the future.

- please describe the filtering method in more detail. It is not clear to me what kind of polynomial was used for the LOESS. Moreover, it is not clear whether the authors used several iteration to get more 'robust' estimates. More important, what is the influence of the LOESS-filtering method on the result, in particular on the linearity of the response.

Reply: The filtering is discussed in more detail now. LOESS is used here only as one low pass filter. We use this to be consistent with Marsicek et al (2018). (Our original motivation is to interpret the millennial variability found in Marsicek et al). Marsicek et al also verified the locally weighted regression (Loess) by generalized additive model (GAMM) fit. We test some of our results simply using running mean and the results remain qualitatively similar.

- please describe the method - used to compute the significance of the correlation – in more detail. If I understand the authors correctly, an AR1 process is only fitted to the ALL-forcing simulation on the different time scales. The Monte-Carlo method is then used to produce an ensemble (PDF) of fitted curves. Then the correlations between the fitted curves and the ALL forcing run are computed and the 95% confidence level is determined afterwards. If I understood the authors correctly, I am
10   wondering if this method is sufficient. I would think that an AR1 process has to be fitted to the ALL forcing run and the superposition (sum of the response of the four individual simulations). Then two ensembles - one for the ALL forcing and one ensemble for the superposition – have to be computed using the Monte-Carlo method. The correlations between these two ensembles have to be used to determine the confidence level. Please make also more clear why you choose the AR1 as a benchmark and how robust the parameter of the AR1 process is, in particular for the orbital time scale.

15   Reply: We think that the randomization on ALL should be sufficient. This is because the key here is to use randomization to destroy the serial relation between ALL and sum. This can be done by randomize either ALL or sum, or both of them. Indeed, we have tested both cases, randomizing one or both time series and confirmed they are the same.
The reviewer is correct in that, strictly speaking, the AR(1) coefficient should be different for each region and should be used for the test of significance. Here, we used the global mean as a common test, mainly for simplicity. Most importantly, our
20   focus here is on the linear response features over the globe, between different regions. Therefore, a common test makes it easy for comparison among different spatial scales and regions. For example, if regional tests are performed, it will be impossible to plot the significance test on the summery figure of Fig.3 and Fig.4 for comparison of different spatial scales. Similarly, it will be hard to compare the value as well as the significance among different regions and spatial scales in Fig.5 and 6. In addition, the global mean AR(1) is meant as a crude representation of most AR(1)'s for different regions. Indeed,
25   except for the orbital scale, the global mean AR(1) is larger than most of the regional AR(1) so that the global mean AR(1) serves as a stricter test. At the orbital scale, the global mean AR(1) is about the middle of the regional AR(1)'s. Finally, we did emphasize that, if one's focus is on a specific region, the regional AR(1) should be used for re-evaluation of the significance. These points are now discussed explicitly in section 2.2.

30   - it is not clear to me why the authors did not do a spectral analysis of the runs like e.g. wavelet analysis, power spectrum, cross power spectrum ...
Reply: Our study is a first preliminary study. Our interests here is mainly on the linear responses on slow time evolution at the orbital and millennial scales. Given only 11,000 years, it is difficult to derive spectral details with high significance. Nevertheless, we agree it will be interesting to explore the spectral features in the future.

- why was the analysis based on the model grid and not on climate modes using e.g. EOF analysis?

Reply: Fixed region is more practical for using model to interpret the real world proxy. Our original motivation is to interpret the regional climate response over North America and Europe as discussed in Marsicek et al (2018). For overall climate response in the model, it is a good idea to perform this in the EOF space.

Minor comments:

- please be more precise (whole text): please rewrite sentences like 'the linear response is strong' => the response is almost linear; the response is similar to that of a linear system

Reply: Thank you for your comments. We have attempted to clarify these terminologies.

- whole text: I would prefer: forcings => forcing factors

Reply: Done!

- page 3, line 8-9: Please rewrite the sentence

Reply: We have deleted it here and explained the data processing in much more details later in 2.2 as follows:

[revised manuscript text omitted]

**Report #1**

**Submitted on 17 May 2019**

**Anonymous Referee #1**

5 I think, the manuscript still suffers from a lack of clarity and conciseness, and the replies to some of the points of my previous review are unsatisfactory or even contradictory. In particular, regarding my previous point

(1) the reply does not explicitly clarify whether the manuscript aims only at my question labelled Q-1 or also Q-2 sometimes, but it seems it is just Q-1, and that the linearity is measured, as before, by the correlation between the SUM and the ALL

10 response, BUT the last sentence of the reply to my previous point

(1) Throughout this work, it seems that the following two *different* questions are mixed up, which makes it basically impossible to evaluate the conclusions drawn from the results:

**Q-1.** How linear is the response to external forcing? If we denote the temperature resonse to the full external forcing,

15 $F_{all}(t) = F_1(t) + F_2(t) + F_3(t) + F_4(t)$, by $T_R(F_{all}(t))$, the response to the individual forcings by $T_R(F_i(t))$ (with $i =$ *1, . . . , 4*), and the internal temperature variability of the five model simulations by $T_{I,all}, T_{I,1}, T_{I,2}, T_{I,3}, T_{I,4}$, respectively, then the linearity of the response could be defined by the extent to which the statement

$$T_R(F_{all}(t)) = \sum_{i=1}^{4} T_R(F_i(t)) \qquad (1)$$

holds, and the linearity could be measured by the correlation between the forced response on the left and that on the right hand side of the above equation. In the manuscript, however, the correlation is computed (see Section 2.2) from the full 'forced plus internal' temperature variability, i.e. between $T_R(F_{all}(t)) + T_{I,all}(t)$ and $\sum_{i=1}^{4} T_R(F_i(t)) + T_{I,i}(t)$, Since this latter correlation is influenced by the signal-to-noise ratio, $Var(T_R)/Var(T_I)$, a small correlation does not necessarily

25 indicate the absence of linearity, because it could be that simply the signal-to-noise ratio is small, although the response is still perfectly linear. (One would need ensembles of model simulations for each of the five forcing scenarios, and then use the ensemble average in order to suppress the internal variability.) Hence, Q-1 cannot be answered by this approach (without additional information), unless one would always obtain correlations close to unity, which would indicate strong linearity.

**Q-2.** What is the relative importance of externally forced vs. internal variability, assuming the response were linear? To

30 answer this question one could use the correlation computed from the full temperature variability, as done in the manuscript, but one had to assume the linearity which, however, is to be proven by this work, in particular, for different temporal and spatial scales.

Hence, the authors should clarify the above issues, and make explicit which of their results contributes to which one of the above two questions. This will also help to clarify the implications of the conclusions for various research fields.

**Reply: We thank the reviewer for this comment again. We apologize for not directly replied to the question in the last reply. In the last reply, section 1 was almost rewritten, largely to address these two questions. In addition, in the previous subsection 2.2, this issue is addressed extensively again. However, we admit that the two questions are not addressed explicitly. In this new revision, we addressed these two questions explicitly in section 2 by adding a new subsection 2.2 Assessment Strategy. Discussions explicitly on the two questions are also made in section 4. (see the revision with tracking).**

(2) contradicts this. Specifically, the reply says that, if the full response is largely dominated by only one forcing such that the other responses are negligible, then this would already imply strong linearity of this response to that single forcing. But if there is only one forcing producing a response, then the employed linearity definition has no meaning anymore! If in that scenario the obtained correlation would be large, it would only mean that the externally forced response dominates over internal variability. But that would be Q-2 again (see above). And it is not obvious to me why the different forcings, which also influence the climate system through different physical mechanisms, should lead to responses of comparable magnitude.

**Reply: Thanks for the comment again. We think we understand your point better now and agree with the reviewer on this. We have added a note in section 4 as follows: "It should also be kept in mind that, if the response is dominated by that to a single forcing, the assessment of linear response here becomes one that is more relevant to the question of the forced response vs internal variability, as discussed in Question 2 in subsection 2.2. As a further step, though, one can examine if the magnitude of the total response responds to the magnitude of this single forcing linearly."**

(2) Even if we had ensembles available for each of the forcing scenarios, it would still be possible to obtain a large correlation coefficient although the response is only weakly linear (i.e., mostly non-linear), if the individual response to, for example, one of the forcings $F_i$ is much larger than the responses to the remaining forcings, because in this case the full temperature variability might still be dominated by the response to the strong forcing (the non-linear interactions might still be relatively small). Thus, one would need to know the strength (e.g., in terms of variance) of the responses to the various individual forcings.

**Reply: See the reply above.**

Regarding my previous point

(4) it is still not clear to me why the variance of the internal variability at millennial and orbital time scales, Var(INT.mil.orb), should be (at least roughly) the same as the variance of the internal variability at decadal and centennial time scales, Var(INT.dec.cen). Whereas Var(INT.dec.cen) is the integral of the power spectrum of the internal variability from approximately 1/(2500yrs) to 1/(20yrs), according to the time scale definitions, Var(INT.mil.orb) is the integral from 1/(11000yrs) to 1/(2500yrs). Why should these two integrals yield roughly the same variance? Or do I misunderstand the employed definition of the SNR at millennial and orbital time scales?

**Reply: This is a good question. Given the limited time series of a single realization, we do not know the variance of internal variability at slow time scale. In Fig.6-7, we estimate the signal/noise by using decadal as noise. This is only for a heuristics attempt to understand the pattern of linearity in different regions. The fact that the signal/noise ratio seems correlated with the linearity may indicate indeed that internal variability at slow time scale may be proportional to that at shorter time scale. This may be possible if we think of a red noise spectrum where the power kept increasing with the time scale, and, therefore, the power at slow time scale may be proportional to that at faster time scales. But this assumes the slope of the power remains the same which we can't justify. In addition, it should be pointed out that in Fig.6-7, we only need $\text{Var}_{orb,mill}(T_{I,all})$ to be proportional, but not equal, to $\text{Var}_{cent+dec}\left(T_{I,all} + T_R(F_{all})\right)$ across different regions. This is a much relaxed condition.**

**Of course, ultimately, the question of internal variability of slow time scales can only be solved using ensemble simulations, we think. A comment is added before the discussion of Fig.6 and 7.**

(4) How is it justified to estimate the variance of the internal variability at orbital and millennial time scales by the full variance at centennial and decadal variability (page 7, last paragraph)? That is, why should we have

$$\text{Var}_{orb,mill}\left(T_{I,all}\right) \approx \text{Var}_{cent+dec}\left(T_{I,all} + T_R(F_{all})\right)? \qquad (2)$$

Even if we assume that $\text{Var}_{cent+dec}\left(T_R(F_{all})\right)$ is small compared to $\text{Var}_{cent+dec}\left(T_{I,all}\right)$, this does not imply anything about the relation between $\text{Var}_{cent+dec}\left(T_{I,all}\right)$ and $\text{Var}_{orb,mill}\left(T_{I,all}\right)$. Maybe it could be helpful to investigate the power spectra of the temperature variability under the various forcing scenarios?

**Reply: See reply above.**

Regarding my previous point

5 (6) I did not find any answer how the AR(1) fit was done. It does probably make a notable difference whether the fit is done using the lag-1 auto-correlation coefficient, or by fitting an exponential to the auto-correlation function, or by fitting an AR(1) power spectrum.

(6) Please, be a bit more explicit how the significance levels are computed. For example, how is the AR(1) fit done in case of
10 the correlation, and what is the bootstrap design for the error index?

**Reply: We thank the reviewer for this comment. The fit is done using the lag-1 auto-correlation coefficient. A comment is added in subsection 2.3.**

I suggest to improve the clarity and conciseness of the questions to be answered, of the definitions employed and of the simplifying assumptions made, in order to bring the conclusions drawn on solid ground.

**Reply: We thank the reviewer for this comment. We have checked the manuscript twice, again.**

[revised manuscript text omitted]